# Wasserstein-type Gaussian Process Regressions for Input Measurement Uncertainty

## Abstract

Gaussian process (GP) regression is widely used for uncertainty quantification, yet the standard formulation assumes noise-free covariates. When inputs are measured with error, this errors-in-variables (EIV) setting can lead to optimistically narrow posterior intervals and biased decisions. We study GP regression under input measurement uncertainty by representing each noisy input as a probability measure and defining covariance through Wasserstein distances between these measures. Building on this perspective, we instantiate a deterministic projected Wasserstein ARD (PWA) kernel whose one-dimensional components admit closed-form expressions and whose product structure yields a scalable, positive-definite kernel on distributions. Unlike latent-input GP models, PWA-based GPs (PWA-GPs) handle input noise without introducing unobserved covariates or Monte Carlo projections, making uncertainty quantification more transparent and robust.

## 1 Introduction

Gaussian Processes (GPs) are core methods for stochastic function approximation in machine learning and surrogate modeling. They are fully defined by a prior mean and a covariance kernel, making them analytically tractable and able to encode function smoothness (Williams and Rasmussen, 2006). GPs output predictive distributions rather than point estimates and quantify uncertainty and confidence. In a Bayesian framework GPs also propagate hyperparameter uncertainty. They have been applied in forward modeling (Deisenroth, 2010; Vinogradska et al., 2016), uncertainty quantification (Tuo and Wang, 2022; Wang, 2021), parameter optimization (Liu et al., 2021; Qiang et al., 2024), and autonomous experimentation (Noack et al., 2019; Stach et al., 2021; Noack et al., 2021; Thomas et al., 2022). Standard GP regression assumes noise-free inputs $X$, modeling observations from $f : \mathbb{R}^d \to \mathbb{R}$ as

$$y = f(X) + \epsilon, \quad \epsilon \sim \mathcal{N}(0, \sigma^2).$$

When the inputs are noisy, i.e., we observed $U = X + \epsilon_X$ as inputs, this "errors-in-variables" (EIV) setting can degrade performance (Zhou et al., 2023). Figure 1 shows how input uncertainty can lead a GP to fail to estimate $f$. Thus, to recover the true function for noisy input, GPs must incorporate input uncertainty or address EIV. The following proposition describes the effect of EIV in terms of the uncertainty quantification of prediction.

The next result provides the technical link between the proposed Wasserstein GP and uncertainty quantification, it shows that a standard GP can undercover when input noise is ignored.

**Proposition 1.** *Consider the errors-in-variables model*

$$Y = f(X + \varepsilon_X) + \varepsilon, \qquad \varepsilon_X \sim \mathcal{N}(0, \Sigma_X), \quad \varepsilon \sim \mathcal{N}(0, \sigma^2),$$

*with $\varepsilon_X \perp \varepsilon$. Let $f(x) = c + w^\top x$ be affine with $w \neq 0$. Define the naive $(1 - \alpha)$ interval that ignores input noise:*

$$I_\alpha := [Y - z_{1-\alpha/2}\sigma, \ Y + z_{1-\alpha/2}\sigma].$$

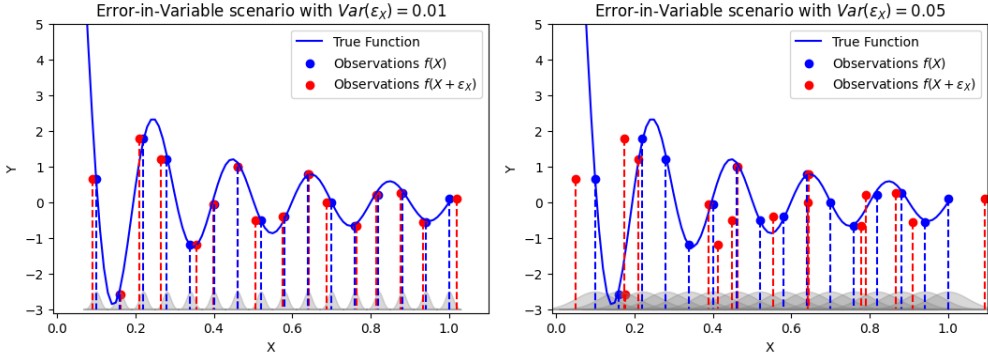

Figure 1: Illustration of error-in-variable regression problem. In both panels, the true function is $y = f(X) = \frac{\sin(10\pi \cdot X)}{2X} + (X-1)^4$, but in each case the "true" input locations $X$ are contaminated with measurement errors (with standard deviation of 0.01 and 0.05 on the left and right, respectively). Using a GP to accurately infer the true function (the blue line) must account for the fact that the input locations $U$ are uncertain.

*Then for any fixed $X$,*

$$\mathbb{P}\big(f(X) \in I_\alpha \mid X\big) = 2\Phi\left(\frac{z_{1-\alpha/2}\sigma}{\sqrt{\sigma^2 + w^\top \Sigma_X w}}\right) - 1 \; < \; 1 - \alpha$$

*whenever $w^\top \Sigma_X w > 0$.*

Proposition 1 formalizes a simple failure mode of a naive GP in an errors-in-variables setting. The interval $I_\alpha$ accounts only for the output noise variance $\sigma^2$, while the true target $f(X) = c + w^\top X$ also varies through the input error term $w^\top \varepsilon_X$, whose variance is $w^\top \Sigma_X w$. As a result, the nominal $(1 - \alpha)$ interval is too narrow and its actual coverage is strictly smaller than $1 - \alpha$. In this paper, we refer to this phenomenon as *undercoverage*.

One approach to taking into account EIV uses latent-variable models: Kennedy and O'Hagan (2001) treat true inputs as latent and integrate them out; Girard et al. (2002) apply this to time series; Damianou and Lawrence (2013) model inputs and outputs jointly; and McHutchon and Rasmussen (2011) convolve the GP prior with an input-noise distribution. These methods capture input uncertainty but require high-dimensional inference, which can be slow or unstable in non-Gaussian settings, and often rely on numerical approximations that may bias results.

Alternatively, one can use kernels on probability measures. If each input is treated as a distribution, for example $U \sim \mathcal{N}(\mu_X, \Sigma_X)$, kernels based on Wasserstein distances compare these inputs directly (see Panaretos and Zemel, 2019). Candelieri et al. (2022) study univariate discrete measures, and Meunier et al. (2022); Bonet et al. (2023) develop sliced Wasserstein kernels for general distributions.

In this work, we view each noisy input $U_i$ as a probability measure $\mu_i$ and place a GP prior directly on the space of such measures. We instantiate this idea with the PWA kernel, which preserves the familiar automatic relevance determination (ARD) structure while encoding input uncertainty through Wasserstein distances between one-dimensional marginals. Intuitively, when the input noise is small, the PWA-GP behaves similarly to a GP fitted on the (latent) noise-free inputs $(X, Y)$, whereas increasing dispersion of the input distributions naturally inflates posterior uncertainty, leading to more honest predictive intervals in error-in-variables problems. Closely related, Paty and Cuturi (2019); Lin et al. (2020) optimize projection directions or subspaces, whereas our proposed PCPWA deliberately fixes its directions and uses them in a separable GP covariance.

In the PWA kernel (6), separate kernel parameters for each input dimension act as ARD-style length scales, handling dimension-specific uncertainty via an automatic relevance determination construction (Williams and Rasmussen, 2006). This avoids full multivariate optimal transport and randomized slicing while still

capturing anisotropic input variability in a way that directly serves uncertainty quantification rather than merely introducing a new kernel form.

In EIV problems, finite-slice implementations of sliced Wasserstein kernels (Meunier et al., 2022) introduce additional Monte Carlo approximation error on top of measurement noise. In contrast, our PWA construction uses deterministic (coordinate or fixed-basis) projections, avoiding slice-induced randomness while retaining valid projected kernels and ARD flexibility. Our theoretical analysis focuses on the case $p = 1$, where covering-number arguments yield uniform error bounds. We summarize related methods in Table A.2, discuss computational complexities in Appendix G, and compare a representative subset in our uncertainty-quantification experiments.

Motivated by previous works on measure-input models (Bachoc et al., 2017; Szabó et al., 2016; Muandet et al., 2017) and and Wasserstein kernels on measures (Kolouri et al., 2016; Meunier et al., 2022; Bonet et al., 2023), we focus on using the measure-input for uncertainty quantification via GP regression, modeling each noisy input as a probability measure and working in the Wasserstein space of distributions. In a simple linear-Gaussian EIV model we show that standard GPs that ignore input noise produce predictive intervals that strictly undercover.

**Summary of our contributions.**

- We develop a deterministic PWA kernel and its principal-component variant PCPWA, which define GPs directly on distributions via products of closed-form one-dimensional Wasserstein kernels. The construction preserves automatic relevance determination (ARD) while avoiding latent variables and stochastic slicing schemes commonly used for error-in-variables GPs.

- We establish a deterministic net-extension bound for the posterior mean of PWA-based GPs and, under a well-specified GP prior, obtain a uniform high-probability posterior band controlled by the posterior standard deviation. This yields a sufficient condition under which a pointwise posterior interval contains the simultaneous band.

- Through experiments on synthetic data, accelerator calibration, and NOAA drifter trajectories, we demonstrate that PWA-GPs provide competitive point predictions and markedly improved uncertainty quantification in terms of coverage and CRPS and relative to standard kernels and sliced Wasserstein baselines under input noise.

## 2 GPs with Wasserstein-type Kernels

### 2.1 Wasserstein and Gromov-Wasserstein distances

Wasserstein distances improve how distributional inputs are handled in machine learning. Arjovsky et al. (2017) highlighted their use for output distributions, but they also apply to input uncertainty.

Intuitively, imagine moving "sand" from a pile at $x$ to fill a hole at $y$: the cost is the mass moved times the distance (Mallasto and Feragen, 2017). We write $\mathcal{P}_p(X)$ for the set of Borel probability measures on $X$ with finite $p$th moment, i.e. $\int d_X(x, x_0)^p \, d\mu(x) < \infty$ for some (equivalently any) $x_0 \in X$. Formally:

**Definition 2.** *(p-Wasserstein distance) Let $p \in [1, \infty)$ and let $\mu, \nu \in \mathcal{P}_p(X)$ be probability measures on a metric space $(X, d_X)$ with finite $p$th moments. The p-Wasserstein distance is*

$$W_p(\mu, \nu) = \left( \inf_{\gamma \in \Gamma(\mu, \nu)} \int_{X \times X} d_X(x, y)^p \, d\gamma(x, y) \right)^{1/p}, \tag{1}$$

*where $\Gamma(\mu, \nu)$ denotes the set of couplings with marginals $\mu$ and $\nu$. The case $p = 1$ is the Earth Mover's Distance (Peyré et al., 2019).*

$W_p(\mu, \nu)$ is the minimal "effort" to transport $\mu$ to $\nu$, raising cost to the $p$th power. These distances scale polynomially in asymptotic regimes (Panaretos and Zemel, 2020; Weed and Bach, 2019).

**Definition 3.** *(Gromov-Wasserstein distance) Let $\mu \in \mathcal{P}_p(X)$ and $\nu \in \mathcal{P}_p(Y)$, where $(X, d_X)$ and $(Y, d_Y)$ are metric spaces. The p-Gromov-Wasserstein distance is*

$$GW_p(\mu, \nu) := \left( \inf_{\pi \in \Gamma(\mu, \nu)} \iint_{(X \times Y)^2} \left| d_X(x, x') - d_Y(y, y') \right|^p d\pi(x, y) \, d\pi(x', y') \right)^{1/p}. \tag{2}$$

$GW_p$ matches the relational geometry of $X$ and $Y$, finding correspondences that minimize distortion.

In essence, $GW_p$ generalizes $W_p$ to compare distributions on different spaces by focusing on pairwise distances. $W_p$ uses a common ground metric, while $GW_p$ handles cross-domain structures, making both valuable for practical tasks in image processing, machine learning, and statistical physics. However, we lack knowledge of the PD property for GW-type kernels.

While the statement of Wasserstein and Gromov-Wasserstein kernels is straightforward, calculating $W_p$ and $GW_p$ can be quite challenging. For $\boldsymbol{x} \in \mathbb{R}^1$, we can have a closed-form expression for any $p \geq 1$:

**Proposition 4.** *(Meunier et al., 2022) Let $\mu, \nu$ be probability measures defined on the real line $\mathbb{R}^1$ and denote their cumulative distribution functions by $F_\mu(x)$ and $F_\nu(x)$. Then,*

$$W_p(\mu, \nu) = \left( \int_0^1 \left| F_\mu^{-1}(q) - F_\nu^{-1}(q) \right|^p dq \right)^{1/p}. \tag{3}$$

## 2.2 Wasserstein-type covariance kernels

We can define $k_{W_p}$ as follows,

$$k_{W_p}(\mu, \nu) = \lambda \exp\left(-\sigma \, W_p(\mu, \nu)^p\right) \tag{4}$$

To use $k_{W_p}$ as a GP covariance function, it is not enough that $W_p$ be a metric. A standard sufficient condition is that the exponent $\psi(\mu, \nu)$ be *conditionally negative definite* (CND): a symmetric function $\psi$ on $\mathcal{P}$ is CND if for every $n \geq 1$, every $\mu_1, \ldots, \mu_n \in \mathcal{P}$, and every coefficients $c_1, \ldots, c_n \in \mathbb{R}$ satisfying $\sum_{i=1}^n c_i = 0$, one has

$$\sum_{i,j=1}^n c_i c_j \psi(\mu_i, \mu_j) \leq 0.$$

By Schoenberg's theorem, if $\psi$ is CND and $\psi(\mu, \mu) = 0$, then $k(\mu, \nu) = \exp(-\sigma \psi(\mu, \nu))$ is positive definite for every $\sigma > 0$.

In (4), the exponent is $\psi(\mu, \nu) = W_p(\mu, \nu)^p$, not $W_p(\mu, \nu)^2$ unless $p = 2$. In general, $W_p(\mu, \nu)^p$ need not be CND on $\mathbb{R}^d$ for arbitrary $d > 1$, so positive definiteness is not automatic outside special settings. This is why we either restrict attention to cases where positive definiteness is known (e.g., certain one-dimensional or Gaussian settings) or use projected/sliced constructions whose one-dimensional factors are positive definite.

A special case is when the inputs are Gaussian and $p = 2$: here the projection is not needed, the kernel (4) is positive definite in the Gaussian setting, and closed forms for $W_2$ enable efficient computation. For example,

$$W_2^2\big(\mathcal{N}(\mu_x, \Sigma_x), \mathcal{N}(\mu_y, \Sigma_y)\big) = \|\mu_x - \mu_y\|^2 + \mathrm{tr}\Big(\Sigma_x + \Sigma_y - 2(\Sigma_x^{1/2} \Sigma_y \Sigma_x^{1/2})^{1/2}\Big).$$

If $\Sigma_x = \Sigma_y$, this reduces to $\|\mu_x - \mu_y\|^2$, so (4) becomes the usual squared-exponential (Gaussian/RBF) kernel on the means (Dowson and Landau, 1982; Peyré et al., 2019; Horn and Johnson, 2012). Alternatively, one may use a *sliced* kernel by averaging over 1D projections $u \in \mathbb{S}^{d-1}$,

$$k(\mu, \nu) = \lambda \exp\big(-\sigma \, \mathbb{E}_u[W_p(\mu_u, \nu_u)^p]\big), \tag{5}$$

which also remains PD (Meunier et al., 2022).

**Model names used in the paper.** We refer to the GP prior with covariance $k_{W_p}$ in (4) as a *Wasserstein GP* (WGP). The GP using the sliced kernel (5) is denoted SWGP. The GP using our projected ARD kernel (6) is denoted PWA-GP, and its fixed-basis/principal-component variant (7) is denoted PCPWA. After these first definitions we use only the abbreviations.

To ensure PD in higher dimensions without solving a $d$-dimensional optimal transportation problem, we use *Projected Wasserstein ARD* kernels. For measures $\mu, \nu$ on $\mathbb{R}^d$, define

$$k(\mu, \nu) = \lambda \prod_{i=1}^{d} \exp\big(-\sigma_i\, W_p(\mu_i, \nu_i)^p\big), \tag{6}$$

where $\mu_i$ is the marginal distribution of measure $\mu$ along the $i$-th dimension defined on $\mathbb{R}^1$; whenever the 1D factors are PD, so is the product kernel (Wilson and Nickisch, 2015; Bilionis et al., 2013). Similar construction can be done using Gromov-Wasserstein distances, yet the projections can be defined through marginals of different dimensions. Unlike previous EIV methods that introduce latent true inputs and require high-dimensional integration or approximate marginalization (Kennedy and O'Hagan, 2001; Girard et al., 2002; Damianou and Lawrence, 2013; McHutchon and Rasmussen, 2011), PWA avoids explicit latent-variable inference.

The PWA construction is a deterministic fixed-projection alternative to sliced Wasserstein kernels. It uses coordinate projections, or a basis fixed once from the training design, and assigns a separate positive distance weight $\sigma_i$ to each selected direction. Thus, it trades rotational invariance and adaptive direction selection for a positive-definite product kernel with closed-form one-dimensional transport terms and ARD-style weighting.

Equation (6) has one global amplitude $\lambda$ and direction-specific distance weights $(\sigma_i)_{i=1}^{d}$. We do not introduce coordinate-specific amplitudes: in a product kernel, only their product would affect the covariance. The weights $\sigma_i$ are ordinary GP hyperparameters that define the weighted geometry of the selected projections; we do not interpret them as separately identifiable physical noise parameters.

Beyond projecting onto coordinate axes, one may also project onto a fixed orthonormal basis of $\mathbb{R}^d$, such as principal components. Let $\{v_r\}_{r=1}^{m} \subset \mathbb{R}^d$ be fixed orthonormal directions and define the linear functional

$$\ell_r : \mathbb{R}^d \to \mathbb{R}, \qquad \ell_r(x) = v_r^\top x.$$

For a Borel probability measure $\mu$ on $\mathbb{R}^d$, define its projected marginal along $v_r$ by

$$\mu_{v_r} := (\ell_r)_{\#}\mu.$$

Equivalently, for every Borel set $A \subset \mathbb{R}$, $\mu_{v_r}(A) = \mu(\{x \in \mathbb{R}^d : v_r^\top x \in A\})$. Here $(\ell_r)_{\#}\mu$ denotes the push-forward of $\mu$ through $\ell_r$. For a Borel probability measure $\mu$ on $\mathbb{R}^d$, we denote by $\mu_{v_r} := (\ell_r)_{\#}\mu$ the push-forward of $\mu$ through $\ell_r$, i.e., the one-dimensional marginal of $\mu$ along direction $v_r$. Explicitly, for every Borel set $A \subset \mathbb{R}$,

$$\mu_{v_r}(A) = ((\ell_r)_{\#}\mu)(A) := \mu(\ell_r^{-1}(A)) = \mu(\{x \in \mathbb{R}^d : v_r^\top x \in A\}).$$

We use the notation $(\ell_r)_{\#}\mu$ rather than $\ell_r(\mu)$ because $\ell_r$ acts on points $x \in \mathbb{R}^d$, while the induced action on measures is the push-forward map.

We define the *principal-component projected Wasserstein ARD* (PCPWA) kernel by

$$k_{\mathrm{PCPWA}}(\mu, \nu) = \lambda \prod_{r=1}^{m} \exp\big(-\sigma_r\, W_p(\mu_{v_r}, \nu_{v_r})^p\big), \tag{7}$$

with direction-specific length scales $(\sigma_r)_{r=1}^{m}$ and a global amplitude $\lambda$.

When $m = d$ and $v_r = e_r$ are the canonical basis vectors, (7) reduces exactly to the coordinate PWA kernel (6), so PWA is a special case of PCPWA. Whenever the 1D factors in (7) are PD, their pullbacks by $\ell_r$ are PD on $\mathcal{P}_p(\mathbb{R}^d)$, and the product over $r$ remains PD by closure under products. Thus, PCPWA inherits the same kernel-validity guarantees as PWA in those validated 1D settings while allowing projections onto any fixed system of directions, including principal components computed once from the design.

### 2.3 Uniform Error Bound for Wasserstein-type GP (p=1)

To assess models under input uncertainty, we derive a uniform bound on the posterior mean over a compact class of 1D distributions. Since our PWA kernel (6) is built from 1D marginals, we limit the theory below to the case $p = 1$. The deterministic net-extension argument parallels Theorem 3.1 of Lederer et al. (2019), while the posterior-probability corollary stated below additionally assumes a well-specified GP prior.

Let $[a, b] \subset \mathbb{R}$ and let $\mathcal{P}$ be a set of Borel probability measures supported on $[a, b]$ such that the Wasserstein covering numbers satisfy $M(\tau, \mathcal{P}) \leq C\tau^{-\alpha_0}$ for some $\alpha_0 > 0$. Assume the quantile functions are uniformly Lipschitz in their argument, i.e., there exists $\ell > 0$ such that

$$|F_\mu^{-1}(q) - F_\mu^{-1}(q')| \leq \ell|q - q'| \qquad \text{for all } q, q' \in [0, 1], \ \mu \in \mathcal{P}.$$

Let $f : \mathcal{P} \to \mathbb{R}$ be $L_f$-Lipschitz with respect to $W_1$. Given observations $(\mu_i, y_i)_{i=1}^N$, define

$$K_{ij} := k(\mu_i, \mu_j), \qquad k_\mu := \big(k(\mu, \mu_1), \ldots, k(\mu, \mu_N)\big)^\top, \qquad \alpha := (K + \sigma_*^2 I_N)^{-1}y.$$

Then

$$\nu_N(\mu) = k_\mu^\top \alpha, \qquad \sigma_N^2(\mu) = k(\mu, \mu) - k_\mu^\top (K + \sigma_*^2 I_N)^{-1} k_\mu.$$

Define

$$L_k := \sup_{\nu \in \mathcal{P}} \sup_{\mu \neq \mu'} \frac{|k(\mu, \nu) - k(\mu', \nu)|}{W_1(\mu, \mu')}, \qquad L_{\nu_N} := N L_k \|\alpha\|_\infty,$$

and let $\omega_{\sigma_N}$ denote a modulus of continuity for $\sigma_N$, i.e.

$$|\sigma_N(\mu) - \sigma_N(\nu)| \leq \omega_{\sigma_N}(\tau) \qquad \text{whenever } W_1(\mu, \nu) \leq \tau.$$

The next theorem is the deterministic extension step that turns control on a finite $\tau$-net into a uniform bound over $\mathcal{P}$.

**Theorem 5.** *Let $\{\bar{\mu}_1, \ldots, \bar{\mu}_M\}$ be a $\tau$-net of $\mathcal{P}$ in $W_1$. Assume $f : \mathcal{P} \to \mathbb{R}$ is $L_f$-Lipschitz with respect to $W_1$, the kernel $k$ is $L_k$-Lipschitz in its first argument with respect to $W_1$, and $\sigma_N$ admits the modulus of continuity $\omega_{\sigma_N}$. If, for some constant $B \geq 0$,*

$$|f(\bar{\mu}_j) - \nu_N(\bar{\mu}_j)| \leq B\,\sigma_N(\bar{\mu}_j), \qquad j = 1, \ldots, M,$$

*then for every $\mu \in \mathcal{P}$,*

$$|f(\mu) - \nu_N(\mu)| \leq B\,\sigma_N(\mu) + (L_f + L_{\nu_N})\tau + B\,\omega_{\sigma_N}(\tau).$$

The proof appears in Appendix D, which mirrors Lederer et al. (2019) but imposes conditions on $L_\infty$ bounds instead of requiring its continuity, which is different from Lederer et al. (2019), and one can use the 1D CDF Lipschitz nets from Proposition 4 to bound $M(\tau, \mathcal{P})$.

The proof appears in Appendix D. It is a deterministic net-extension argument: the only stochastic ingredient enters later, when the finite-net inequalities are obtained from Gaussian posterior tails under a well-specified GP model.

**Corollary 6.** *Assume in addition that the regression model is well specified: $f \sim \mathcal{GP}(0, k)$, the design points $(\mu_i)_{i=1}^N$ are fixed, and the observations satisfy*

$$y_i = f(\mu_i) + \epsilon_i, \qquad \epsilon_i \overset{\text{ind}}{\sim} \mathcal{N}(0, \sigma_*^2), \qquad i = 1, \ldots, N.$$

*Then for any $\tau > 0$ and $\delta \in (0, 1)$, conditional on $\mathcal{D}_N$, with posterior probability at least $1 - \delta$,*

$$|f(\mu) - \nu_N(\mu)| \leq \sqrt{\beta(\tau)}\,\sigma_N(\mu) + \gamma(\tau) \qquad \text{for all } \mu \in \mathcal{P},$$

*where*

$$\beta(\tau) = \left[\Phi^{-1}\left(1 - \frac{\delta}{2\,M(\tau, \mathcal{P})}\right)\right]^2, \qquad \gamma(\tau) = (L_f + L_{\nu_N})\,\tau + \sqrt{\beta(\tau)}\,\omega_{\sigma_N}(\tau).$$

The same argument applies to the PCPWA kernel with $D$ replacing $W_1$.

**Corollary 7.** *Let $\mathcal{P}$ be a set of Borel probability measures on $\mathbb{R}^d$, and define*

$$D(\mu,\nu) := \left( \sum_{r=1}^{m} a_r \, W_1(\mu_{v_r}, \nu_{v_r})^2 \right)^{1/2}$$

*for fixed directions $\{v_r\}_{r=1}^m \subset \mathbb{R}^d$ and weights $a_r > 0$. Assume $f : \mathcal{P} \to \mathbb{R}$ is $L_f$-Lipschitz with respect to $D$, the PCPWA kernel $k_{\mathrm{PCPWA}}$ from (7) is $L_k$-Lipschitz in its first argument with respect to $D$, and $\sigma_N$ admits the modulus of continuity $\omega_{\sigma_N}$ with respect to $D$. If $(\mathcal{P}, D)$ admits a $\tau$-net of size $M_{\mathrm{PCPWA}}(\tau, \mathcal{P})$ and the model is well specified with prior $f \sim \mathcal{GP}(0, k_{\mathrm{PCPWA}})$, then for any $\tau > 0$ and $\delta \in (0,1)$, conditional on $\mathcal{D}_N$, with posterior probability at least $1 - \delta$,*

$$|f(\mu) - \nu_N(\mu)| \le \sqrt{\beta_{\mathrm{PCPWA}}(\tau)} \, \sigma_N(\mu) + \gamma_{\mathrm{PCPWA}}(\tau) \qquad \text{for all } \mu \in \mathcal{P},$$

*where*

$$\beta_{\mathrm{PCPWA}}(\tau) = \left[ \Phi^{-1}\left( 1 - \frac{\delta}{2\,M_{\mathrm{PCPWA}}(\tau, \mathcal{P})} \right) \right]^2, \qquad \gamma_{\mathrm{PCPWA}}(\tau) = (L_f + L_{\nu_N})\tau + \sqrt{\beta_{\mathrm{PCPWA}}(\tau)} \, \omega_{\sigma_N}(\tau).$$

The preceding generic covering-number argument is stated for $p = 1$. For the $p = 2$ PCPWA kernels used in our experiments, Appendix F gives a kernel-specific specialization based on the $L^2$ quantile embedding. It makes explicit how the ARD weights $(\sigma_r)_{r=1}^m$, the number of fixed projections $m$, and the chosen directions $(v_r)_{r=1}^m$ enter the metric, projected covering number, and posterior-band width. Beyond this quantile-embedding $p = 2$ specialization, sharper entropy bounds for particular distribution classes are left for future work.

The preceding corollaries are net-based posterior band statements analogous in spirit to the Wasserstein-space analysis of Meunier et al. (2022), but here the finite-net step is made explicit and the deterministic extension from the net to the full input class is separated from the model-based Gaussian tail argument. This immediately implies the following proposition, which shows how the band width can prevent the undercoverage highlighted in Proposition 1.

**Proposition 8.** *Assume the posterior event in Corollary 6 holds with parameters $(\tau, \delta)$, i.e., conditional on $\mathcal{D}_N$,*

$$|f(\mu) - \nu_N(\mu)| \le \sqrt{\beta(\tau)} \, \sigma_N(\mu) + \gamma(\tau) \quad \text{for all } \mu \in \mathcal{P}$$

*with posterior probability at least $1 - \delta$. Fix $\alpha \in (0,1)$ and $z := z_{1-\alpha/2}$. If*

$$z > \sqrt{\beta(\tau)} \qquad \text{and} \qquad \sigma_N(\mu_X) \ge \frac{\gamma(\tau)}{z - \sqrt{\beta(\tau)}},$$

*then*

$$\mathbb{P}\big( f(\mu_X) \in \big[ \nu_N(\mu_X) \pm z\, \sigma_N(\mu_X) \big] \,\big|\, \mathcal{D}_N \big) \ge 1 - \delta.$$

Thus, on the stated simultaneous-band event, the variance condition ensures interval containment. This is a posterior containment statement under the well-specified model; it does not by itself establish frequentist EIV calibration or conservativeness at the nominal level.

Sliced Wasserstein kernels (5) are positive definite, but in practice are implemented using a finite number of random projections, which introduces Monte Carlo approximation error. By contrast, PWA/PCPWA are deterministic projections and avoid this additional randomness.

**Proposition 9.** *Fix a direction $u \in \mathbb{S}^{d-1}$ and assume the analogue of Corollary 6 holds for the 1D projected model with parameters $(\tau, \delta)$: conditional on $\mathcal{D}_N$,*

$$|f(\mu) - \nu_N^u(\mu)| \le \sqrt{\beta_u(\tau)} \, \sigma_N^u(\mu) + \gamma_u(\tau) \quad \text{for all } \mu \in \mathcal{P}$$

*with posterior probability at least $1 - \delta$. Fix $\alpha \in (0,1)$ and $z := z_{1-\alpha/2}$. If*

$$z > \sqrt{\beta_u(\tau)} \qquad \text{and} \qquad \sigma_N^u(\mu_X) \ge \frac{\gamma_u(\tau)}{z - \sqrt{\beta_u(\tau)}},$$

*then*

$$\mathbb{P}\big( f(\mu_X) \in \big[ \nu_N^u(\mu_X) \pm z\, \sigma_N^u(\mu_X) \big] \,\big|\, \mathcal{D}_N \big) \ge 1 - \delta.$$

# 3 Experiments and Applications

Practitioners who model input uncertainty with Gaussian summaries often use a covariance-only input map $x \mapsto \Sigma_x$, so that each input location $x$ is represented as $X \sim \mathcal{N}(\mu_x, \Sigma_x)$ and only the covariance $\Sigma_x$ enters the kernel (Moreno-Muñoz et al., 2018). This approach is efficient when noise is well-approximated by local Gaussian perturbations but cannot represent skew, multimodality, or complex support geometry. The recent work of Song and Han (2023) also tackles input uncertainty by modeling in the space of covariates. Since their low-rank linear framework is not directly comparable to the non-linear GP models studied here, we do not include it in the empirical comparison. We cannot use histogram discretizations like the previous Wasserstein GP literature (Meunier et al., 2022; Bachoc et al., 2017), since the bin-width selection has a major effect on the subsequent model. We evaluate our proposed PWA-GP (and PCPWA where noted) on synthetic and real datasets using: (1) RMSE for posterior-mean prediction accuracy; (2) CRPS (Arnold et al., 2024), which jointly assesses predictive precision and sharpness, and (3) the probability coverage, which measures how well the posterior uncertainty covers the measured test observations. Appendix B contains the definitions of these metrics. Unless otherwise stated, all empirical Wasserstein distances in Tables 2 and 1 use $p = 2$; the $p = 1$ restriction applies only to the formal uniform-error theory in Section 2.3. We optimize all GP hyperparameters by maximizing the log marginal likelihood on the training split, using the same tuning budget across methods: ten random initializations in log scale, L-BFGS-B optimization, positivity constraints for variance and length-scale parameters, and the best converged objective value retained. The length-scale and kernel-scale search ranges are initialized from the empirical pairwise-distance scale of the corresponding representation (means for Euclidean GPs, one-dimensional Wasserstein distances for PWA/PCPWA/SWGP, full Wasserstein distances for WGP, and kernel-mean/MMD distances for KME/MMD); the observation-noise variance is initialized from the sample variance of the responses. For SWGP, we use a fixed set of random projection directions for each run and declare a run numerically divergent when Cholesky factorization fails after jittering or when the predictive RMSE/CRPS exceeds $10^5$; the values reported in the tables are the resulting held-out metrics under this same optimization protocol.

## 3.1 Simulated distributional regression experiments

We study several synthetic distributional regression problems designed to probe errors-in-variables (EIV) behavior and uncertainty quantification. In all cases, each covariate is a finite cloud $U_i = \{u_{ij}\}_{j=1}^{n_i} \subset \mathbb{R}^d$ with associated response $Y_i$, and the task is to predict $Y$ for new clouds. We report test RMSE, empirical coverage of nominal 90% predictive intervals, and CRPS for Regular GP on empirical means, Aggregated GP on individual samples, Wasserstein GPs (WGP, SWGP, PWA-GP, PCPWA), uncertain-input GP UI-GP Girard et al. (2002), and KME/MMD distributional GPs (Meunier et al., 2022).

**1D scenarios.** We consider three one-dimensional settings with Gaussian output noise $\eta_i \sim \mathcal{N}(0, 0.05^2)$. (i) **1D-EIV**: latent covariates $x_i$ are evenly spaced on $[0.05, 0.95]$. We observe $U_i = \{x_i + \varepsilon_{ij}\}_{j=1}^{n_i}$ with heteroscedastic Gaussian measurement noise $\varepsilon_{ij} \sim \mathcal{N}(0, \sigma_i^2)$, where $\sigma_i^2$ increases with $x_i$, and set $Y_i = f(x_i) + \eta_i$ for a nonlinear oscillatory $f$. This is a canonical errors-in-variables problem. (ii) **1D-Var**: $U_i$ consists of $u_{ij} \sim \mathcal{N}(\mu_i, \sigma_i^2)$ with varying variances, and the response depends on both mean and spread, $Y_i = \sin(2\pi\mu_i) + 0.5\sigma_i^2 + \eta_i$. (iii) **1D-Skew**: inputs are log-normal, $u_{ij} \sim \exp(\mathcal{N}(m_i, s_i^2))$, and the response depends on an inter-quantile range and location, $Y_i = [Q_{0.8}(U_i) - Q_{0.2}(U_i)] + 0.3\sin(m_i) + \eta_i$, where $Q_q(U_i)$ denotes the $q$-quantile of the input distribution (or, equivalently in the simulations, of the empirical cloud $U_i$). 1D-Var and 1D-Skew thus require sensitivity to higher-order distributional features.

**2D scenarios.** We next consider two two-dimensional problems. (iv) **2D-mean** (Gaussian location functional): for each $i$ we draw $\mu_i \in [0.1, 1]$ and sample $u_{ij} \sim \mathcal{N}(\mu_i \mathbf{1}_2, \mathrm{diag}(\sigma_{i1}^2, \sigma_{i2}^2))$ with moderate variances. The response is a smooth function of the mean of $U_i$, averaged over coordinates,

$$Y_i = \frac{1}{n_i} \sum_{j=1}^{n_i} \left[ \sin(u_{ij}) + 2e^{u_{ij}} \right] + \eta_i,$$

so the problem is essentially a Gaussian location model. (v) **2D-aniso-PC** (anisotropic rotated subspace): latent scalars $z_i \in [0.05, 0.95]$ generate samples in a rotated basis $(z_{ij}^{\parallel}, z_{ij}^{\perp})$ via $z_{ij}^{\parallel} = z_i + \epsilon_{ij}^{\parallel}$, $z_{ij}^{\perp} = \epsilon_{ij}^{\perp}$ with $\epsilon_{ij}^{\parallel} \sim \mathcal{N}(0, \sigma_{\parallel}^2(z_i))$ and $\epsilon_{ij}^{\perp} \sim \mathcal{N}(0, \sigma_{\perp}^2)$, where $\sigma_{\parallel}^2(z_i)$ increases with $z_i$ and $\sigma_{\perp}^2$ is small. A fixed $45°$ rotation $R$ maps $u_{ij} = R[z_{ij}^{\parallel}, z_{ij}^{\perp}]^{\top}$, and the response depends only on $z_i$, $Y_i = \sin(4\pi z_i) + 0.5z_i + \eta_i$. Here the intrinsic variation lies along a rotated one-dimensional subspace, so PCA-based kernels such as PCPWA are expected to help.

**High-dimensional scenarios.** Finally, we construct high-dimensional problems from the Ackley function. (vi) **HD-Ackley-5D** and (vii) **HD-Ackley-10D**: latent covariates $x_i \in [-2, 2]^d$ with $d \in \{5, 10\}$ are drawn uniformly, we observe noisy clouds $u_{ij} = x_i + \varepsilon_{ij}$ with $\varepsilon_{ij} \sim \mathcal{N}(0, 0.1^2 I_d)$, and set $Y_i = f_{\text{Ackley}}(x_i) + \eta_i$. These scenarios test how methods scale with dimension in the presence of input noise and nontrivial curvature.

Table 2 reports the resulting RMSE, coverage, and CRPS. In 1D-EIV, Regular GP achieves RMSE 0.309 but under-covers (65%), Aggregated GP is both inaccurate and grossly over-confident (5% coverage), while WGP, PWA-GP, PCPWA, and SWGP all attain similar RMSE ($\approx 0.30$) with coverage $\approx 0.97$ and the lowest CRPS ($\approx 0.17$), matching our EIV coverage analysis. In 1D-Var and 1D-Skew, mean- or moment-based baselines (Regular, UI-GP, KME/MMD) either lose accuracy or under-cover, whereas Wasserstein kernels maintain competitive RMSE and near-nominal coverage. In 1D-Var the Regular GP has the best RMSE/CRPS, but the Wasserstein methods are within about 10-15% in RMSE; in 1D-Skew, MMD-GP attains the lowest RMSE and CRPS but with coverage 0.58, whereas the Wasserstein models trade a slightly higher RMSE for better-calibrated intervals.

In the benign 2D-mean scenario, uncertain-input and KME/MMD GPs-whose assumptions match the Gaussian location structure-achieve the smallest RMSE (0.0004-0.0026) with near-nominal coverage and CRPS, while Wasserstein kernels remain reasonable but conservative (e.g. PWA-GP RMSE 0.0462, coverage 1.0). This shows that our approach does not artificially dominate when the problem is effectively Euclidean since WGP becomes regular GP when $\sigma_{i1} = \sigma_1$ and $\sigma_{i2} = \sigma_2$. By contrast, in the anisotropic 2D-aniso-PC setting, methods based on Gaussian moments or mean embeddings deteriorate markedly (UI-GP and KME-GP RMSE $\approx 0.69$, coverage 0.15), whereas WGP, PWA-GP, PCPWA, and MMD-GP all keep RMSE around 0.20 and coverage between 0.65 and 0.98. PCPWA in particular combines good accuracy (RMSE 0.2036, very close to the best 0.1972) with high coverage (0.95) and low CRPS, benefiting from aligning the kernel with principal directions of the empirical clouds.

The high-dimensional Ackley experiments further probe robustness under complex geometry. In 5D, Wasserstein kernels (especially PWA-GP and WGP) achieve lower RMSE and CRPS than Regular and UI-GP while maintaining higher coverage, and Aggregated GP again fails badly. In 10D, all methods become challenged, and MMD-GP attains the best RMSE/CRPS with reasonably high coverage; nevertheless, Wasserstein kernels remain very close in RMSE (e.g. PWA-GP and WGP within roughly 10% of the best) and retain the second best coverage.

Across all scenarios, sliced-WGP (SWGP) is competitive in 1D but its optimization is numerically unstable in several higher-dimensional cases, as indicated by the extremely large RMSE/CRPS values in Table 2. Overall, PWA-GP and PCPWA consistently deliver well-calibrated predictive uncertainty in EIV settings while retaining competitive RMSE, and they remain robust as the input distributions become heteroscedastic, skewed, anisotropic, and high-dimensional, even when they are not the single best-performing method in raw RMSE.

## 3.2  Calibration of particle accelerators

Particle accelerators play a central role in scientific discovery and industrial applications. In a typical machine, focusing elements such as quadrupole magnets guide the charged particle beam and preserve beam quality, measured by the transverse emittance at the end of the accelerator. Here we study the relationship between the strengths of five quadrupole magnets in a linac section and the resulting beam emittance.

For each of the $N = 500$ machine settings we observe a five-dimensional vector of magnet strengths and two emittances (horizontal and vertical). Each magnet is specified with a relative tolerance (5% or 0.5%) around

its nominal strength Tao et al. (2017), so the "input" available to a regression model is a noisy proxy $U_i$ of an unknown true setting $X_i$-a classical errors-in-variables situation. We therefore model each setting as

$$U_i \sim \mathcal{N}(\mu_i, \Sigma_i), \qquad \Sigma_i = \text{diag}(\mu_i \odot \varepsilon),$$

where $\mu_i \in \mathbb{R}^5$ is the magnet vector for shot $i$ and $\varepsilon \in \{0.05, 0.005\}$ is the relative error. From each $\mathcal{N}(\mu_i, \Sigma_i)$ we draw a small cloud of samples to represent $U_i$, and define the scalar response $Y_i = (\text{horizontal emittance}) \times (\text{vertical emittance})/10^{-12}$. We use an 80/20 train-test split.

We fit both Euclidean and distributional GP models. Euclidean baselines are standard GPs on empirical means $m_i = \mathbb{E}[U_i]$ with RBF, Matérn 3/2, Matérn 5/2 and exponential kernels. Distributional competitors are our Wasserstein GPs (PWA-GP, WGP, PCPWA) that act on the sample clouds using separable, full and PCA-based Wasserstein kernels, together with the uncertain-input GP UI-GP of Girard et al. (2002).

Table 1 reports the results for both 5% and 0.5% relative errors. All methods (except SWGP) nearly interpolate the training data (train RMSE $< 2 \times 10^{-2}$, not shown). On the test set, UI-GP consistently achieves the best accuracy and calibration: RMSE 0.1340 with coverage 0.86 and CRPS 0.0557 at 5% error, improving to RMSE 0.1294, coverage 0.92 and CRPS 0.0441 at 0.5% error-very close to the nominal 90% target. Euclidean GPs on means have larger RMSE (0.22-0.24 at 5%, 0.22-0.28 at 0.5%) and higher CRPS, but still decent coverage (0.91-0.98), reflecting that in this nearly Gaussian, low-dimensional setting a moment-based EIV correction already works well.

Our Wasserstein GPs (PWA-GP, WGP, PCPWA) are broadly comparable to the mean-based Euclidean GPs on this dataset: they achieve similar qualitative behavior and conservative coverage (0.96-0.98), but do not provide a systematic improvement in point accuracy or CRPS. This is consistent with the structure of the input uncertainty here, each $U_i$ is close to Gaussian and its dispersion is largely determined by the nominal setting $\mu_i$ via a fixed relative tolerance. Therefore, much of the useful information is already captured by the empirical mean representation. The SWGP is numerically unstable for both error levels, with diverging RMSE and CRPS and zero coverage, and is therefore not a viable competitor on this dataset.

Overall, the accelerator results align with our simulation results in Table 2 (1D-EIV): when the input noise is close to Gaussian and the response depends mainly on low-order moments, mean-based and uncertain-input GPs can match (and in this case outperform) Wasserstein GPs, whereas the latter show clear benefits in the more non-Gaussian and anisotropic EIV scenarios of Section 3.1.

| Method | 5% error | | | 0.5% error | | |
|---|---|---|---|---|---|---|
| | RMSE | Cov. | CRPS | RMSE | Cov. | CRPS |
| RBF GP | 0.2186 | 0.910 | 0.0883 | 0.2177 | 0.920 | 0.0806 |
| Matérn 3/2 GP | 0.2351 | 0.930 | 0.1010 | 0.2378 | 0.930 | 0.0997 |
| Matérn 5/2 GP | 0.2228 | 0.940 | 0.0912 | 0.2242 | 0.940 | 0.0892 |
| Exp. GP | 0.2432 | 0.980 | 0.1380 | 0.2847 | 0.980 | 0.1392 |
| PWA-GP | 0.2798 | 0.980 | 0.1690 | 0.3373 | 0.980 | 0.1681 |
| SWGP | $2.7234{\times}10^6$ | 0.000 | $2.1404{\times}10^6$ | $3.1142{\times}10^6$ | 0.000 | $3.9275{\times}10^6$ |
| WGP | 0.3169 | 0.970 | 0.1949 | 0.2806 | 0.970 | 0.1401 |
| PCPWA | 0.3072 | 0.960 | 0.1682 | 0.2828 | 0.970 | 0.1430 |
| Uncertain-input GP | 0.1340 | 0.860 | 0.0557 | 0.1294 | 0.920 | 0.0441 |

Table 1: Accelerator dataset: test RMSE, empirical coverage of nominal 90% predictive intervals (Cov.), and CRPS for two relative error levels. Euclidean GPs use empirical means as inputs; distributional GPs act on full input distributions.

### 3.3 Noisy Trajectories from NOAA Drifters

This experiment compares GP-RBF and WGP-RBF in predicting mean temperatures, emphasizing input uncertainty. We evaluate models using RMSE and CRPS metrics on NOAA's Global Drifter Program dataset, containing hourly drifter trajectories and sea surface temperatures (SST) observations (Elipot et al., 2016;

2022). While the SST measurements are collected over space at a high temporal frequency, scientific interest is often focused on the time-aggregated behavior of SSTs to understand large-scale modes of ocean variability (Philander, 1983; Mantua and Hare, 2002; Schlesinger and Ramankutty, 1994; Thompson and Wallace, 2000). Therefore, we consider these trajectories as distributional input models, where the distribution of inputs represents the collection of geospatial positions corresponding to the hourly measurements. The data is preprocessed by selecting trajectories with low temperature variance ($Y_{\text{var\_temp}}$). Each collection of scattered points on a trajectory is then represented by a 2D normal distribution with empirical mean position and empirical covariance matrix. We have to emphasize that the trajectories of drifters (i.e., floating buoys on the oceanic surface) are highly noisy, self-intersecting and therefore usual functional data representation is not appropriate here. Therefore, we consider these trajectories as distributional input models, where both measurement noise and ocean variability-information GP-RBF ignores by using only point estimates. The response ($Y_{\text{mean\_temp}}$) is the mean SST. We train on one year's data (e.g., 1999) and test on another year (e.g., 1998), initially focusing on 100 trajectories with lowest variance to ensure valid scalar responses.

We then evaluated all eight methods on every pair of training/testing years from 1987 to 2021 using the same reduced-space NOAA setup as above. The sliced Wasserstein GP is omitted here because of its much higher memory cost on this benchmark. Figure 2 reveals three consistent patterns. First, every method exhibits a dark diagonal or near-diagonal band, indicating that prediction is easiest when the training and testing years are the same or nearby. Several distributional models, especially WGP, PWA-GP, and PCPWA, nearly interpolate the diagonal cells, reflecting that once the within-year spatial distributions are well matched, the mean temperature response is easy to recover.

Second, the dominant failure mode is not symmetric in time. The hardest regime is training on earlier years and testing on later years, which appears as the upper-left block of the heatmaps. For Regular GP, the median RMSE increases from 4.71 when training on newer years and testing on older years to 12.87 in the reverse direction; the corresponding CRPS increases from 2.84 to 6.69. The same directional asymmetry is present for the Wasserstein models, but the transition is visually smoother for the projected kernels: PWA-GP changes from median RMSE 5.37 to 10.09, while full WGP changes from 6.58 to 10.56. This suggests a genuine temporal domain shift in the NOAA drifter distributions, with later years harder to predict from earlier-year training data.

Third, once all methods are compared, the NOAA benchmark does not support a simple "WGP is uniformly best" conclusion. Instead, PWA-GP gives the **strongest overall transfer map**, attaining the lowest RMSE on 357 off-diagonal year pairs and the lowest CRPS on 409 off-diagonal pairs, compared with 42 and 45 for full WGP. MMD-GP is also competitive, with 260 off-diagonal RMSE wins and 242 off-diagonal CRPS wins. Full WGP and PCPWA remain attractive from an uncertainty-quantification standpoint, however: their average empirical coverage on this benchmark is 0.73 and 0.73, respectively, compared with 0.66 for Regular GP. Aggregated GP overfits the diagonal most aggressively but degrades sharply off-diagonal. Overall, the expanded NOAA experiment shows that distribution-aware kernels still offer a clear advantage for temporal transfer, but on this real dataset the most reliable choices are the projected Wasserstein kernel PWA-GP and the strong non-OT baseline MMD-GP, rather than full WGP alone.

## 4  Discussion

We have argued that uncertainty quantification, rather than purely pointwise accuracy, should guide the design of GP models when inputs are measured with error. By lifting inputs to probability measures and endowing this space with Wasserstein geometry, PWA-based GPs provide a simple, deterministic alternative to latent-input and sampling-based EIV approaches. The theory now separates a deterministic net-extension bound from a model-based GP posterior statement: under Lipschitz and covering-number conditions, one obtains a uniform high-probability posterior band and a simple sufficient condition for conservative credible intervals. Our experiments confirm that these ideas are consistent with substantially better-calibrated predictive intervals on real data. Our empirical findings suggest that classical Euclidean kernels are often overly confident when covariates are noisy, whereas Wasserstein-type kernels and PWA in particular produce credible intervals that track the true variability across different noise regimes and non-Gaussian input distributions. This makes PWA-GP especially appealing in scientific and engineering applications, such

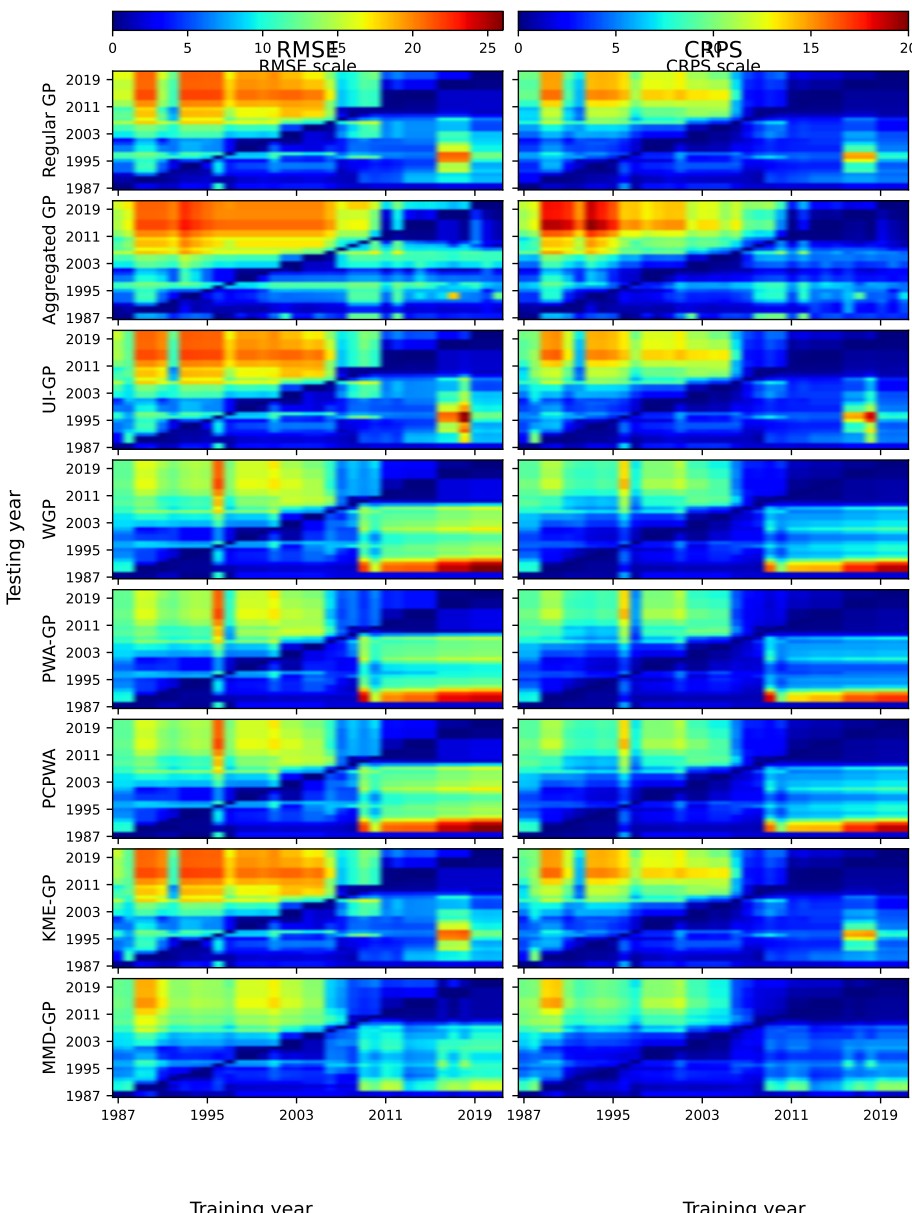

Figure 2: NOAA drifter benchmark across all training/testing year pairs (1987–2021). Each row corresponds to one method and shows RMSE (left column) and CRPS (right column) heatmaps. Color scales are shared within each metric and clipped at the 95th percentile for readability. Across methods, the strongest errors occur under cross-year transfer rather than on the diagonal, and the temporal shift is directional: training on earlier years and testing on later years is substantially harder than the reverse. Among the methods compared here, PWA-GP produces the broadest region of low RMSE/CRPS and attains the largest number of best year-pair scores, while MMD-GP is the strongest non-OT competitor.

as accelerator calibration and oceanographic monitoring, where decisions are driven by the reliability of uncertainty quantification.

Future work includes extending the analysis to high-dimensional, non-Gaussian noise (Tekriwal et al., 2026) and non-Euclidean input spaces (Luo et al., 2026), incorporating learned or problem-specific transport costs.

| Scenario | Metric | Reg | Agg | WGP | SWGP | PWA | PCPWA | UI | KME | MMD |
|---|---|---|---|---|---|---|---|---|---|---|
| 2D-mean | RMSE | 8.500e-03 | 1.061e-01 | 5.280e-02 | 1.015e+07 | 4.620e-02 | 1.615e-01 | 4.000e-04 | 2.100e-03 | 2.600e-03 |
| | Cov. | 4.250e-01 | 1.000e-01 | **1.000e+00** | 0.000e+00 | 6.250e-01 | **9.750e-01** | **1.000e+00** | **1.000e+00** | **9.750e-01** |
| | CRPS | 5.100e-03 | 5.940e-02 | 4.540e-02 | 7.752e+06 | 3.970e-02 | 1.108e-01 | 2.000e-03 | 8.900e-03 | 1.010e-02 |
| 1D-EIV | RMSE | 3.090e-01 | 5.712e-01 | 3.014e-01 | 2.995e-01 | 3.014e-01 | 3.014e-01 | 3.462e-01 | 4.179e-01 | 3.029e-01 |
| | Cov. | 6.500e-01 | 5.000e-02 | **9.670e-01** | **9.670e-01** | **9.670e-01** | **9.670e-01** | 5.670e-01 | 5.330e-01 | 6.670e-01 |
| | CRPS | 1.848e-01 | 4.791e-01 | 1.716e-01 | 1.722e-01 | 1.716e-01 | 1.716e-01 | 2.179e-01 | 2.844e-01 | 1.775e-01 |
| 1D-Var | RMSE | 1.612e-01 | 4.810e-01 | 1.849e-01 | 1.819e-01 | 1.849e-01 | 1.849e-01 | 2.016e-01 | 2.697e-01 | 1.743e-01 |
| | Cov. | 7.670e-01 | 0.000e+00 | **9.830e-01** | **9.830e-01** | **9.830e-01** | **9.830e-01** | 6.330e-01 | 6.170e-01 | 7.670e-01 |
| | CRPS | 9.150e-02 | 3.648e-01 | 1.070e-01 | 1.058e-01 | 1.070e-01 | 1.070e-01 | 1.135e-01 | 1.552e-01 | 9.900e-02 |
| 1D-Skew | RMSE | 1.768e-01 | 4.779e-01 | 1.629e-01 | 1.633e-01 | 1.629e-01 | 1.629e-01 | 1.762e-01 | 1.892e-01 | 1.406e-01 |
| | Cov. | 4.330e-01 | 0.000e+00 | **9.500e-01** | **9.670e-01** | **9.500e-01** | **9.500e-01** | 4.170e-01 | 4.500e-01 | 5.830e-01 |
| | CRPS | 1.075e-01 | 3.618e-01 | 9.730e-02 | 9.750e-02 | 9.730e-02 | 9.730e-02 | 1.057e-01 | 1.152e-01 | 8.000e-02 |
| 2D-aniso-PC | RMSE | 1.972e-01 | 5.435e-01 | 2.086e-01 | 2.541e+06 | 2.130e-01 | 2.036e-01 | 6.893e-01 | 6.893e-01 | 2.009e-01 |
| | Cov. | 6.750e-01 | 2.500e-02 | **9.750e-01** | 0.000e+00 | **9.500e-01** | **9.500e-01** | 1.500e-01 | 1.500e-01 | 6.500e-01 |
| | CRPS | 1.086e-01 | 3.846e-01 | 1.252e-01 | 2.581e+06 | 1.211e-01 | 1.124e-01 | 5.528e-01 | 5.528e-01 | 1.098e-01 |
| HD-Ackley-5D | RMSE | 6.071e-01 | 2.483e+00 | 5.413e-01 | 4.080e+08 | 5.190e-01 | 5.562e-01 | 6.062e-01 | 6.574e-01 | 5.080e-01 |
| | Cov. | 7.750e-01 | 2.500e-02 | 9.000e-01 | 0.000e+00 | 8.870e-01 | **9.630e-01** | 7.750e-01 | 7.750e-01 | 7.870e-01 |
| | CRPS | 3.501e-01 | 1.879e+00 | 3.302e-01 | 3.081e+07 | 2.715e-01 | 3.147e-01 | 3.495e-01 | 3.607e-01 | 2.969e-01 |
| HD-Ackley-10D | RMSE | 5.925e-01 | 2.487e+00 | 5.873e-01 | 5.888e+00 | 6.003e-01 | 6.110e-01 | 5.962e-01 | 6.112e-01 | 5.450e-01 |
| | Cov. | 2.630e-01 | 7.500e-02 | 6.880e-01 | 0.000e+00 | 7.000e-01 | 7.620e-01 | 6.750e-01 | 6.250e-01 | 7.630e-01 |
| | CRPS | 4.138e-01 | 1.887e+00 | 3.492e-01 | 2.431e+00 | 3.567e-01 | 3.718e-01 | 3.470e-01 | 3.552e-01 | 3.309e-01 |

Table 2: Simulated distributional regression experiments. For each scenario we report test RMSE, empirical coverage of nominal 90% predictive intervals (Cov.), and CRPS. "Reg" = Regular GP on empirical means, "Agg" = Aggregated GP, "UI" = uncertain-input GP. Extremely large SWGP values indicate numerical divergence. Values of Cov. above 0.9 are shown in **bold**. The execution time is reported in Appendix H

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

## A    Related Tables

## B    Evaluation metrics

We report three complementary metrics for predictive performance. Let $\widehat{m}_i$ and $\widehat{s}_i$ denote the posterior mean and posterior standard deviation for the $i$th test input, and let $y_i$ be the corresponding test response.

First, the root mean squared error (RMSE) measures point-prediction accuracy:

$$\text{RMSE} = \left( \frac{1}{n_{\text{test}}} \sum_{i=1}^{n_{\text{test}}} (\widehat{m}_i - y_i)^2 \right)^{1/2}.$$

Second, for a nominal $100(1-\alpha)\%$ predictive interval

$$C_i = \left[ \widehat{m}_i - z_{1-\alpha/2}\widehat{s}_i, \ \widehat{m}_i + z_{1-\alpha/2}\widehat{s}_i \right],$$

the empirical coverage is

$$\text{Cov}_{1-\alpha} = \frac{1}{n_{\text{test}}} \sum_{i=1}^{n_{\text{test}}} \mathbf{1}\{y_i \in C_i\}.$$

Coverage below the nominal level is called *undercoverage* and indicates that the predictive intervals are too narrow, i.e., the model is overconfident.

Third, we report the continuous ranked probability score (CRPS), a proper scoring rule for the full predictive distribution:

$$\text{CRPS}(F_i, y_i) = \int_{-\infty}^{\infty} \left( F_i(t) - \mathbf{1}\{y_i \le t\} \right)^2 dt,$$

where $F_i$ is the predictive cumulative distribution function for the $i$th test point. Lower CRPS is better; it rewards both calibration and sharpness.

## C    Proof for Proposition 1

*Proof.* With $f(x) = c + w^\top x$, we have

$$Y = f(X + \varepsilon_X) + \varepsilon = f(X) + w^\top \varepsilon_X + \varepsilon.$$

Thus $f(X) - Y = -(w^\top \varepsilon_X + \varepsilon)$ and conditional on $X$,

$$w^\top \varepsilon_X + \varepsilon \sim \mathcal{N}(0, \ w^\top \Sigma_X w + \sigma^2).$$

Therefore,

$$\mathbb{P}(f(X) \in I_\alpha \mid X) = \mathbb{P}\left( |w^\top \varepsilon_X + \varepsilon| \le z_{1-\alpha/2}\sigma \right) = 2\Phi\left( \frac{z_{1-\alpha/2}\sigma}{\sqrt{\sigma^2 + w^\top \Sigma_X w}} \right) - 1,$$

which is strictly less than $1 - \alpha$ whenever $w^\top \Sigma_X w > 0$. $\qquad\square$

## D    Proof for Theorem 5

*Proof.* We first verify the Lipschitz continuity of $\nu_N$. Since

$$\nu_N(\mu) = \sum_{i=1}^{N} \alpha_i k(\mu, \mu_i),$$

and $k$ is $L_k$-Lipschitz in its first argument with respect to $W_1$,

$$|\nu_N(\mu) - \nu_N(\mu')| \le \sum_{i=1}^{N} |\alpha_i|\, |k(\mu, \mu_i) - k(\mu', \mu_i)| \le L_k \Big(\sum_{i=1}^{N} |\alpha_i|\Big) W_1(\mu, \mu').$$

Because $\alpha = (K + \sigma_*^2 I_N)^{-1} y$ and $K + \sigma_*^2 I_N$ is invertible, $\|\alpha\|_\infty < \infty$, hence

$$\sum_{i=1}^{N} |\alpha_i| \le N\|\alpha\|_\infty \quad \text{and therefore} \quad |\nu_N(\mu) - \nu_N(\mu')| \le L_{\nu_N} W_1(\mu, \mu').$$

Now let $k_\mu = (k(\mu, \mu_1), \ldots, k(\mu, \mu_N))^\top$ and set $A = (K + \sigma_*^2 I_N)^{-1}$. Then

$$\sigma_N^2(\mu) = k(\mu, \mu) - k_\mu^\top A k_\mu.$$

By assumption, $\sigma_N$ has modulus of continuity $\omega_{\sigma_N}$, i.e.

$$|\sigma_N(\mu) - \sigma_N(\mu')| \le \omega_{\sigma_N}(\tau) \qquad \text{whenever } W_1(\mu, \mu') \le \tau.$$

Let $\{\bar{\mu}_1, \ldots, \bar{\mu}_M\}$ be a $\tau$-net of $\mathcal{P}$. Fix any $\mu \in \mathcal{P}$ and choose $j$ such that $W_1(\mu, \bar{\mu}_j) \le \tau$. Then

$$|f(\mu) - \nu_N(\mu)| \le |f(\mu) - f(\bar{\mu}_j)| + |f(\bar{\mu}_j) - \nu_N(\bar{\mu}_j)| + |\nu_N(\bar{\mu}_j) - \nu_N(\mu)|.$$

The first and third terms are bounded by Lipschitz continuity:

$$|f(\mu) - f(\bar{\mu}_j)| \le L_f \tau, \qquad |\nu_N(\bar{\mu}_j) - \nu_N(\mu)| \le L_{\nu_N} \tau.$$

By hypothesis,

$$|f(\bar{\mu}_j) - \nu_N(\bar{\mu}_j)| \le B\, \sigma_N(\bar{\mu}_j) \le B\, [\sigma_N(\mu) + \omega_{\sigma_N}(\tau)].$$

Combining the three bounds yields

$$|f(\mu) - \nu_N(\mu)| \le B\, \sigma_N(\mu) + (L_f + L_{\nu_N})\tau + B\, \omega_{\sigma_N}(\tau),$$

as claimed. $\qquad\square$

*Proof of Corollary 6.* Let $\{\bar{\mu}_1, \ldots, \bar{\mu}_M\}$ be a $\tau$-net of $\mathcal{P}$ in $W_1$, where $M = M(\tau, \mathcal{P})$. Under the well-specified GP model, conditional on $\mathcal{D}_N$, for each $j = 1, \ldots, M$,

$$f(\bar{\mu}_j) \mid \mathcal{D}_N \sim \mathcal{N}\big(\nu_N(\bar{\mu}_j), \sigma_N^2(\bar{\mu}_j)\big).$$

Fix $j$. If $\sigma_N(\bar{\mu}_j) > 0$, then

$$Z_j := \frac{f(\bar{\mu}_j) - \nu_N(\bar{\mu}_j)}{\sigma_N(\bar{\mu}_j)} \mid \mathcal{D}_N \sim \mathcal{N}(0, 1),$$

and therefore, by the definition

$$\beta(\tau) = \left[\Phi^{-1}\Big(1 - \frac{\delta}{2M(\tau, \mathcal{P})}\Big)\right]^2,$$

we have

$$\mathbb{P}\Big(|f(\bar{\mu}_j) - \nu_N(\bar{\mu}_j)| > \sqrt{\beta(\tau)}\, \sigma_N(\bar{\mu}_j) \,\Big|\, \mathcal{D}_N\Big) = \mathbb{P}\Big(|Z_j| > \Phi^{-1}\Big(1 - \frac{\delta}{2M(\tau, \mathcal{P})}\Big)\Big)$$

$$= \frac{\delta}{M(\tau, \mathcal{P})}.$$

If instead $\sigma_N(\bar{\mu}_j) = 0$, then the posterior law is degenerate and $f(\bar{\mu}_j) = \nu_N(\bar{\mu}_j)$ almost surely given $\mathcal{D}_N$, so the same inequality holds with probability $0 \le \delta/M(\tau, \mathcal{P})$.

Hence, for every $j = 1, \ldots, M$,

$$\mathbb{P}\Big(|f(\bar{\mu}_j) - \nu_N(\bar{\mu}_j)| > \sqrt{\beta(\tau)}\,\sigma_N(\bar{\mu}_j)\,\Big|\,\mathcal{D}_N\Big) \leq \frac{\delta}{M(\tau, \mathcal{P})}.$$

Applying the union bound over the $M(\tau, \mathcal{P})$ net points yields

$$\mathbb{P}\Big(|f(\bar{\mu}_j) - \nu_N(\bar{\mu}_j)| \leq \sqrt{\beta(\tau)}\,\sigma_N(\bar{\mu}_j)\text{ for all } j = 1, \ldots, M\,\Big|\,\mathcal{D}_N\Big) \geq 1 - \delta.$$

On this event, Theorem 5 applies with $B = \sqrt{\beta(\tau)}$, giving

$$|f(\mu) - \nu_N(\mu)| \leq \sqrt{\beta(\tau)}\,\sigma_N(\mu) + (L_f + L_{\nu_N})\tau + \sqrt{\beta(\tau)}\,\omega_{\sigma_N}(\tau) \qquad \text{for all } \mu \in \mathcal{P}.$$

Since

$$\gamma(\tau) = (L_f + L_{\nu_N})\tau + \sqrt{\beta(\tau)}\,\omega_{\sigma_N}(\tau),$$

the claimed bound follows. $\qquad\qquad\square$

*Proof of Corollary 7.* Let $\{\bar{\mu}_1, \ldots, \bar{\mu}_M\}$ be a $\tau$-net of $(\mathcal{P}, D)$, where $M = M_{\mathrm{PCPWA}}(\tau, \mathcal{P})$. Under the well-specified GP model with prior $f \sim GP(0, k_{\mathrm{PCPWA}})$, conditional on $\mathcal{D}_N$, for each $j = 1, \ldots, M$,

$$f(\bar{\mu}_j) \mid \mathcal{D}_N \sim \mathcal{N}\big(\nu_N(\bar{\mu}_j),\, \sigma_N^2(\bar{\mu}_j)\big).$$

Define

$$\beta_{\mathrm{PCPWA}}(\tau) = \left[\Phi^{-1}\Big(1 - \frac{\delta}{2M_{\mathrm{PCPWA}}(\tau, \mathcal{P})}\Big)\right]^2.$$

Arguing exactly as in the proof of Corollary 6, for each net point $\bar{\mu}_j$,

$$\mathbb{P}\Big(|f(\bar{\mu}_j) - \nu_N(\bar{\mu}_j)| > \sqrt{\beta_{\mathrm{PCPWA}}(\tau)}\,\sigma_N(\bar{\mu}_j)\,\Big|\,\mathcal{D}_N\Big) \leq \frac{\delta}{M_{\mathrm{PCPWA}}(\tau, \mathcal{P})}.$$

A union bound therefore gives, conditional on $\mathcal{D}_N$,

$$|f(\bar{\mu}_j) - \nu_N(\bar{\mu}_j)| \leq \sqrt{\beta_{\mathrm{PCPWA}}(\tau)}\,\sigma_N(\bar{\mu}_j), \qquad j = 1, \ldots, M,$$

with posterior probability at least $1 - \delta$.

Now apply Theorem 5 with the pseudo-metric $D$ in place of $W_1$, using the assumed $D$-Lipschitz bounds for $f$ and $k_{\mathrm{PCPWA}}$ and the modulus of continuity $\omega_{\sigma_N}$ with respect to $D$, with

$$B = \sqrt{\beta_{\mathrm{PCPWA}}(\tau)}.$$

This yields

$$|f(\mu) - \nu_N(\mu)| \leq \sqrt{\beta_{\mathrm{PCPWA}}(\tau)}\,\sigma_N(\mu) + (L_f + L_{\nu_N})\tau + \sqrt{\beta_{\mathrm{PCPWA}}(\tau)}\,\omega_{\sigma_N}(\tau) \qquad \text{for all } \mu \in \mathcal{P},$$

with posterior probability at least $1 - \delta$. Writing

$$\gamma_{\mathrm{PCPWA}}(\tau) = (L_f + L_{\nu_N})\tau + \sqrt{\beta_{\mathrm{PCPWA}}(\tau)}\,\omega_{\sigma_N}(\tau)$$

completes the proof. $\qquad\qquad\square$

# E   Proof for Proposition 9

*Proof.* Fix $u \in \mathbb{S}^{d-1}$. Let $\mathcal{E}_u$ denote the posterior event that the projected uniform bound holds:

$$|f(\mu) - \nu_N^u(\mu)| \le \sqrt{\beta_u(\tau)}\, \sigma_N^u(\mu) + \gamma_u(\tau) \qquad \text{for all } \mu \in \mathcal{P}.$$

By assumption, $\mathbb{P}(\mathcal{E}_u \mid \mathcal{D}_N) \ge 1 - \delta$.

On $\mathcal{E}_u$, evaluating at $\mu = \mu_X$ gives

$$|f(\mu_X) - \nu_N^u(\mu_X)| \le \sqrt{\beta_u(\tau)}\, \sigma_N^u(\mu_X) + \gamma_u(\tau).$$

Assume $z > \sqrt{\beta_u(\tau)}$ and $\sigma_N^u(\mu_X) \ge \gamma_u(\tau)/(z - \sqrt{\beta_u(\tau)})$. Then

$$\sqrt{\beta_u(\tau)}\, \sigma_N^u(\mu_X) + \gamma_u(\tau) \le z\, \sigma_N^u(\mu_X),$$

so $|f(\mu_X) - \nu_N^u(\mu_X)| \le z\, \sigma_N^u(\mu_X)$, which is equivalent to $f(\mu_X) \in I_\alpha^u(\mu_X)$. Therefore,

$$\mathbb{P}(f(\mu_X) \in I_\alpha^u(\mu_X) \mid \mathcal{D}_N) \ge \mathbb{P}(\mathcal{E}_u \mid \mathcal{D}_N) \ge 1 - \delta,$$

which proves the claim. □

# F   Kernel-specific $p = 2$ consequences for PCPWA

This appendix specializes the generic net-extension argument to the $p = 2$ PCPWA kernel used in the experiments. All kernel hyperparameters are treated as fixed, as in Corollaries 6 and 7.

**Corollary 10.** *Let $\mathcal{P} \subset \mathcal{P}_2(\mathbb{R}^d)$, let $\{v_r\}_{r=1}^m \subset \mathbb{R}^d$ be fixed orthonormal directions, and let $\lambda > 0$ and $\sigma_r > 0$, and $\sigma_*^2 > 0$. Consider the $p = 2$ PCPWA kernel*

$$k_{\mathrm{PCPWA},2}(\mu, \nu) = \lambda \exp\left( -\sum_{r=1}^m \sigma_r W_2(\mu_{v_r}, \nu_{v_r})^2 \right).$$

*Define the projected quantile metric*

$$\rho_\sigma(\mu, \nu) := \left( \sum_{r=1}^m \sigma_r W_2(\mu_{v_r}, \nu_{v_r})^2 \right)^{1/2}.$$

*Assume that $f : \mathcal{P} \to \mathbb{R}$ is $L_f$-Lipschitz with respect to $\rho_\sigma$, that the design points are fixed, and that*

$$f \sim \mathcal{GP}(0, k_{\mathrm{PCPWA},2}), \qquad y_i = f(\mu_i) + \epsilon_i, \qquad \epsilon_i \overset{\mathrm{ind}}{\sim} \mathcal{N}(0, \sigma_*^2).$$

*For a chosen $\tau > 0$ and $\delta \in (0, 1)$, assume $M_{\rho_\sigma}(\tau, \mathcal{P}) < \infty$. Let $L_f \ge 0$, $\eta \in [0, 1 - \delta)$, and define the posterior Lipschitz event*

$$\mathcal{L}_{L_f} := \left\{ \sup_{\rho_\sigma(\mu, \nu) > 0} \frac{|f(\mu) - f(\nu)|}{\rho_\sigma(\mu, \nu)} \le L_f \right\}.$$

*Thus, on $\mathcal{L}_{L_f}$, $f$ is constant on $\rho_\sigma$-zero equivalence classes. Assume that $\mathcal{L}_{L_f}$ is measurable and that, conditional on $\mathcal{D}_N$,*

$$\mathbb{P}\big( \mathcal{L}_{L_f} \mid \mathcal{D}_N \big) \ge 1 - \eta.$$

*Let*

$$\beta_\sigma(\tau) := \left[ \Phi^{-1}\left( 1 - \frac{\delta}{2M_{\rho_\sigma}(\tau, \mathcal{P})} \right) \right]^2.$$

*Further define*

$$c_0 := \sqrt{\frac{2}{e}}, \qquad s_0 := \left( \frac{\lambda \sigma_*^2}{N\lambda + \sigma_*^2} \right)^{1/2},$$

$$L_{\nu,\sigma} := N\lambda c_0 \|\alpha\|_\infty, \qquad L_{s,\sigma} := \frac{N\lambda^2 c_0}{\sigma_*^2 s_0},$$

and

$$\gamma_\sigma(\tau) := \left[ L_f + L_{\nu,\sigma} + \sqrt{\beta_\sigma(\tau)}\, L_{s,\sigma} \right] \tau.$$

Then, conditional on $\mathcal{D}_N$, with posterior probability at least $1 - \delta - \eta$,

$$|f(\mu) - \nu_N(\mu)| \leq \sqrt{\beta_\sigma(\tau)}\, \sigma_N(\mu) + \gamma_\sigma(\tau) \qquad \text{for all } \mu \in \mathcal{P}.$$

When $m = d$ and $v_r = e_r$, this gives the corresponding $p = 2$ PWA result.

*Proof.* By Proposition 4, for every $r$,

$$W_2(\mu_{v_r}, \nu_{v_r})^2 = \int_0^1 \left| F_{\mu_{v_r}}^{-1}(q) - F_{\nu_{v_r}}^{-1}(q) \right|^2 dq.$$

Hence the map

$$T_\sigma(\mu) := \left( \sqrt{\sigma_r}\, F_{\mu_{v_r}}^{-1} \right)_{r=1}^m \in \bigoplus_{r=1}^m L^2([0,1])$$

satisfies

$$\rho_\sigma(\mu, \nu) = \|T_\sigma(\mu) - T_\sigma(\nu)\|.$$

Therefore,

$$k_{\mathrm{PCPWA},2}(\mu, \nu) = \lambda \exp\left( -\|T_\sigma(\mu) - T_\sigma(\nu)\|^2 \right),$$

which is positive definite.

The function $t \mapsto e^{-t^2}$ is $c_0$-Lipschitz on $[0, \infty)$, where $c_0 = \sqrt{2/e}$. The reverse triangle inequality therefore gives

$$|k_{\mathrm{PCPWA},2}(\mu, \nu) - k_{\mathrm{PCPWA},2}(\mu', \nu)| \leq \lambda c_0\, \rho_\sigma(\mu, \mu').$$

Thus the kernel Lipschitz constant may be taken as

$$L_k = \lambda c_0,$$

and hence

$$|\nu_N(\mu) - \nu_N(\mu')| \leq N\lambda c_0 \|\alpha\|_\infty\, \rho_\sigma(\mu, \mu').$$

It remains to obtain an explicit modulus for $\sigma_N$. Let

$$A := (K + \sigma_*^2 I_N)^{-1}, \qquad b_\mu := k_\mu.$$

Since $0 \leq k_{\mathrm{PCPWA},2}(\mu, \nu) \leq \lambda$,

$$\|b_\mu\|_2 \leq \lambda\sqrt{N}, \qquad \|b_\mu - b_{\mu'}\|_2 \leq \lambda c_0\sqrt{N}\, \rho_\sigma(\mu, \mu').$$

Consequently,

$$\left| b_\mu^\top A b_\mu - b_{\mu'}^\top A b_{\mu'} \right| \leq \frac{2N\lambda^2 c_0}{\sigma_*^2} \rho_\sigma(\mu, \mu').$$

Let $\phi_\mu$ be a feature map for $k_{\mathrm{PCPWA},2}$ and define

$$S := \sum_{i=1}^N \phi_{\mu_i} \otimes \phi_{\mu_i}.$$

Then

$$\sigma_N^2(\mu) = \sigma_*^2 \left\langle \phi_\mu, (S + \sigma_*^2 I)^{-1} \phi_\mu \right\rangle.$$

Since $\|\phi_\mu\|^2 = \lambda$ and

$$\|S\|_{\mathrm{op}} \leq \mathrm{tr}(S) = \sum_{i=1}^{N} \|\phi_{\mu_i}\|^2 = N\lambda,$$

we obtain

$$\sigma_N^2(\mu) \geq \frac{\lambda \sigma_*^2}{N\lambda + \sigma_*^2} = s_0^2.$$

Therefore,

$$|\sigma_N(\mu) - \sigma_N(\mu')| \leq \frac{N\lambda^2 c_0}{\sigma_*^2 s_0} \rho_\sigma(\mu, \mu') = L_{s,\sigma} \rho_\sigma(\mu, \mu').$$

Let $\{\bar{\mu}_1, \ldots, \bar{\mu}_M\}$ be a $\tau$-net of $\mathcal{P}$ in $\rho_\sigma$, where $M = M_{\rho_\sigma}(\tau, \mathcal{P})$, and let

$$\mathcal{E}_\tau := \left\{ |f(\bar{\mu}_j) - \nu_N(\bar{\mu}_j)| \leq \sqrt{\beta_\sigma(\tau)} \, \sigma_N(\bar{\mu}_j), \quad j = 1, \ldots, M \right\}.$$

Conditional on $\mathcal{D}_N$, Gaussian posterior tails and a union bound give $\mathbb{P}(\mathcal{E}_\tau \mid \mathcal{D}_N) \geq 1 - \delta$. On $\mathcal{E}_\tau \cap \mathcal{L}_{L_f}$, the deterministic net-extension argument of Theorem 5 applies with $\rho_\sigma$ in place of $W_1$ and with the explicit constants above. The claim follows from the union bound. $\square$

**Corollary 11** (Projected entropy bound and ARD dependence). *Under the assumptions of Corollary 10, define*

$$\mathcal{P}_r := \{\mu_{v_r} : \mu \in \mathcal{P}\} \subset \mathcal{P}_2(\mathbb{R}).$$

*Suppose that, for each $r = 1, \ldots, m$, there exist constants $C_r > 0$, $\alpha_r > 0$, and $\varepsilon_{0,r} > 0$ such that*

$$M_{W_2}(\varepsilon, \mathcal{P}_r) \leq C_r \varepsilon^{-\alpha_r}, \qquad 0 < \varepsilon \leq \varepsilon_{0,r}.$$

*Let*

$$A_{\mathrm{proj}} := \sum_{r=1}^{m} \alpha_r, \qquad C_{\mathrm{proj},\sigma} := \prod_{r=1}^{m} C_r \left(2\sqrt{m\sigma_r}\right)^{\alpha_r}.$$

*For*

$$0 < \tau \leq 2 \min_{1 \leq r \leq m} \sqrt{m\sigma_r} \, \varepsilon_{0,r},$$

*we have*

$$M_{\rho_\sigma}(\tau, \mathcal{P}) \leq C_{\mathrm{proj},\sigma} \, \tau^{-A_{\mathrm{proj}}}.$$

*Consequently, for any $\tau$ satisfying*

$$0 < \tau \leq 2 \min_{1 \leq r \leq m} \sqrt{m\sigma_r} \, \varepsilon_{0,r} \quad and \quad C_{\mathrm{proj},\sigma} \tau^{-A_{\mathrm{proj}}} \geq 1,$$

*define*

$$\bar{\beta}_\sigma(\tau) := \left[ \Phi^{-1}\left(1 - \frac{\delta \tau^{A_{\mathrm{proj}}}}{2 C_{\mathrm{proj},\sigma}}\right) \right]^2$$

*and*

$$\bar{\gamma}_\sigma(\tau) := \left[ L_f + L_{\nu,\sigma} + \sqrt{\bar{\beta}_\sigma(\tau)} \, L_{s,\sigma} \right] \tau.$$

*Then, conditional on $\mathcal{D}_N$, with posterior probability at least $1 - \delta - \eta$,*

$$|f(\mu) - \nu_N(\mu)| \leq \sqrt{\bar{\beta}_\sigma(\tau)} \, \sigma_N(\mu) + \bar{\gamma}_\sigma(\tau) \qquad \text{for all } \mu \in \mathcal{P}.$$

*Proof.* For each $r$, choose an $\varepsilon_r$-net of $\mathcal{P}_r$ in $W_2$, where

$$\varepsilon_r := \frac{\tau}{2\sqrt{m\sigma_r}}.$$

The restriction on $\tau$ ensures that $\varepsilon_r \leq \varepsilon_{0,r}$ for every $r$, so the assumed entropy bound applies to these nets. Partition $\mathcal{P}$ according to the tuple of projected-net elements to which $(\mu_{v_1}, \ldots, \mu_{v_m})$ is assigned, and select one representative measure from each nonempty cell. If $\mu$ and $\mu'$ belong to the same cell, then for every $r$,

$$W_2(\mu_{v_r}, \mu'_{v_r}) \leq 2\varepsilon_r.$$

Therefore,

$$\rho_\sigma(\mu, \mu')^2 \leq \sum_{r=1}^{m} \sigma_r (2\varepsilon_r)^2 = \tau^2.$$

The number of nonempty cells is at most

$$\prod_{r=1}^{m} M_{W_2}\left(\frac{\tau}{2\sqrt{m\sigma_r}}, \mathcal{P}_r\right),$$

which gives

$$M_{\rho_\sigma}(\tau, \mathcal{P}) \leq \prod_{r=1}^{m} C_r \left(\frac{\tau}{2\sqrt{m\sigma_r}}\right)^{-\alpha_r} = C_{\text{proj},\sigma}\, \tau^{-A_{\text{proj}}}.$$

The stated posterior band follows from Corollary 10, since the displayed entropy bound implies

$$\beta_\sigma(\tau) \leq \overline{\beta}_\sigma(\tau).$$

$\square$

## G  Computational complexity of competing methods

We briefly compare the leading-order computational complexity of the GP methods considered in this work. For all models, once a kernel matrix $K \in \mathbb{R}^{N \times N}$ has been assembled, exact GP training via Cholesky decomposition costs $\mathcal{O}(N^3)$, and prediction for $N_\star$ test inputs costs $\mathcal{O}(N^2 + NN_\star)$ (or $\mathcal{O}(NN_\star)$ once the Cholesky factor is cached). The main differences between methods therefore arise from (i) the number of *effective* training points they use and (ii) the cost of computing each kernel entry from the underlying clouds $U_i$.

Throughout, $U_i = \{u_{ij}\}_{j=1}^{m_i} \subset \mathbb{R}^d$ denotes the empirical input distribution for training example $i$, and $M = \max_i m_i$. We assume $m_i \asymp M$ for simplicity.

**Regular GP on means.**  The regular GP baseline (Williams and Rasmussen, 2006) treats each distribution as a single point in $\mathbb{R}^d$ via its empirical mean $\bar{u}_i = m_i^{-1} \sum_{j=1}^{m_i} u_{ij}$. Computing all means costs $\mathcal{O}(NMd)$ once, and a kernel evaluation between two means is $\mathcal{O}(d)$. Building $K$ therefore costs $\mathcal{O}(N^2 d)$, after which GP training is $\mathcal{O}(N^3)$. This is the reference complexity for a standard Euclidean GP with $N$ inputs of dimension $d$.

**Aggregated GP on raw samples.**  The aggregated GP baseline (an ensemble of standard GPs (Williams and Rasmussen, 2006)) treats each sample $u_{ij}$ as an independent input. The effective number of training points becomes $\tilde{N} = \sum_i m_i \asymp NM$. Kernel evaluations remain $\mathcal{O}(d)$, but the kernel matrix is now of size $\tilde{N} \times \tilde{N}$, so training scales as

$$\mathcal{O}(\tilde{N}^3) = \mathcal{O}(N^3 M^3),$$

with kernel assembly cost $\mathcal{O}(N^2 M^2 d)$. This cubic dependence on $M$ makes the aggregated GP rapidly infeasible as the number of samples per distribution grows.

**Uncertain-input GP.**  The uncertain-input GP of Girard et al. (2002) (see also McHutchon and Rasmussen (2011)) replaces each $U_i$ by a Gaussian $\mathcal{N}(\mu_i, \Sigma_i)$, where $\mu_i$ and $\Sigma_i$ are the empirical mean and covariance. Estimating these parameters costs $\mathcal{O}(NMd^2)$ in general (or $\mathcal{O}(NMd)$ for diagonal covariances). The kernel between two Gaussians has a closed form involving $(\Sigma_i + \Sigma_j)$ and $(\mu_i - \mu_j)$; with full covariances this requires an inversion and determinant per pair, for a cost of $\mathcal{O}(d^3)$ per kernel entry and $\mathcal{O}(N^2 d^3)$ to assemble $K$. For diagonal or low-rank covariances the per-pair cost reduces to $\mathcal{O}(d)$ or $\mathcal{O}(kd)$, giving $\mathcal{O}(N^2 d)$ kernel assembly and again $\mathcal{O}(N^3)$ training.

**Full Wasserstein GP.** Wasserstein GPs build kernels from optimal-transport distances between empirical distributions (Mallasto and Feragen, 2017; Bachoc et al., 2017; Panaretos and Zemel, 2019; Candelieri et al., 2022). Computing the pairwise cost matrix between $U_i$ and $U_j$ is $\mathcal{O}(M^2 d)$, and solving the OT problem (e.g. with Sinkhorn or network simplex) requires at least $\mathcal{O}(M^2)$ per iteration and $\mathcal{O}(M^2)$ memory (Peyré et al., 2019). In practice this yields a per-pair distance cost of order $\mathcal{O}(M^2 d)$ to $\mathcal{O}(M^3)$, and thus a total cost of

$$\mathcal{O}(N^2 M^2 d) \quad \text{to} \quad \mathcal{O}(N^2 M^3)$$

for computing $K$, plus the usual $\mathcal{O}(N^3)$ for GP training. This quadratic (or worse) dependence on $M$ motivates the separable and sliced variants below.

**PWA-GP (separable Wasserstein kernel).** Our PWA-GP uses a separable kernel built from one-dimensional Wasserstein distances along each coordinate, still rooted in the OT geometry of Panaretos and Zemel (2019). In 1D, the $W_2$ distance between empirical distributions of size $M$ reduces to sorting, which costs $\mathcal{O}(M \log M)$ per marginal. For $d$ dimensions, a naive implementation therefore costs $\mathcal{O}(dM \log M)$ per pair and $\mathcal{O}(N^2 dM \log M)$ in total to assemble $K$, which is much cheaper than full OT as soon as $M$ is moderately large. Training again costs $\mathcal{O}(N^3)$.

**PCPWA (PCA-based separable Wasserstein kernel).** PCPWA first computes a low-rank PCA basis for the pooled samples and then applies the separable 1D Wasserstein construction along the leading $k$ principal directions. The up-front PCA on all samples costs $\mathcal{O}(NMd^2)$ (or less with randomized SVD), but this is a one-time cost. Thereafter, the per-pair distance uses only $k$ one-dimensional marginals, for a cost $\mathcal{O}(kM \log M)$ per pair and $\mathcal{O}(N^2 kM \log M)$ in total. Since typically $k \ll d$, PCPWA can be substantially cheaper than PWA-GP in high dimensions while also adapting to the intrinsic low-dimensional structure of the clouds.

**Sliced Wasserstein GP.** The sliced Wasserstein GP (Meunier et al., 2022) draws $R$ random projection directions $\{\theta_r\}_{r=1}^R \subset \mathbb{S}^{d-1}$ and approximates the Wasserstein distance by averaging 1D Wasserstein distances along these projections. Projecting all samples in $U_i$ onto $\theta_r$ costs $\mathcal{O}(Md)$ per direction and $\mathcal{O}(RMd)$ in total per cloud. These projections can be precomputed and reused across pairs. Given the projected samples, computing 1D distances between $U_i$ and $U_j$ across all $R$ directions costs $\mathcal{O}(RM \log M)$ per pair. The overall kernel assembly cost is therefore

$$\mathcal{O}(NRMd) \quad \text{(precomputation)} \; + \; \mathcal{O}(N^2 RM \log M) \quad \text{(pairwise distances)},$$

followed by $\mathcal{O}(N^3)$ training. In our code $R$ is treated as a moderate constant, so the dominant dependence is $\mathcal{O}(N^2 M \log M)$.

**KME and MMD GPs.** The KME and MMD kernels embed each empirical distribution into an RKHS using sample averages of feature maps (Muandet et al., 2017; Szabó et al., 2016). With a Gaussian base kernel and empirical distributions of size $M$, the kernel between $U_i$ and $U_j$ typically uses a double sum over samples,

$$k_{\mathrm{KME}}(U_i, U_j) \; \approx \; \frac{1}{M^2} \sum_{a=1}^M \sum_{b=1}^M \exp\left(-\frac{\|u_{ia} - u_{jb}\|^2}{2\ell^2}\right),$$

and similarly for the squared MMD. Each kernel evaluation is therefore $\mathcal{O}(M^2 d)$, and assembling $K$ costs $\mathcal{O}(N^2 M^2 d)$, again followed by $\mathcal{O}(N^3)$ training. Random features can reduce this to $\mathcal{O}(N^2 Dd)$ with $D$ features, but in our experiments we use the exact empirical embedding.

**Low-rank Fréchet EIV regression.** The low-rank Fréchet errors-in-variables (EIV) regression of Song and Han (2023) represents each distributional covariate via a rank-$r$ approximation in a suitable Hilbert space. Constructing these low-rank features from $M$ samples per distribution costs roughly $\mathcal{O}(NMr)$, after which Fréchet regression (or GP regression on the $r$-dimensional coefficients) operates on $N$ inputs of dimension $r$. Kernel assembly is therefore $\mathcal{O}(N^2 r)$, comparable to a regular GP in $r$ dimensions, with the usual $\mathcal{O}(N^3)$

training cost. When $r \ll d$ and $M$ is moderate, this yields a competitive complexity to our PCA-based PCPWA construction.

Table G.1 summarizes the leading-order complexity of the main models as a function of the number of clouds $N$, samples per cloud $M$, and dimension $d$ (ignoring the shared $\mathcal{O}(N^3)$ GP training cost). Regular and uncertain-input GPs (Williams and Rasmussen, 2006; Girard et al., 2002; McHutchon and Rasmussen, 2011) are cheapest in terms of $M$, since they compress each distribution to a finite-dimensional summary. Aggregated GP is prohibitively expensive as soon as $M$ grows. Among fully distributional methods, our separable Wasserstein kernels (PWA-GP, PCPWA), together with SWGP (Meunier et al., 2022), enjoy *linear* or near-linear dependence on $M$ (up to logarithmic factors) thanks to one-dimensional Wasserstein computations (Panaretos and Zemel, 2019; Peyré et al., 2019). In contrast, full WGP, KME, and MMD kernels (Mallasto and Feragen, 2017; Bachoc et al., 2017; Muandet et al., 2017; Szabó et al., 2016) scale at least quadratically in $M$. This helps explain the empirical observation in our simulated and accelerator experiments that PWA-GP and PCPWA provide competitive errors-in-variables performance and well-calibrated uncertainty while remaining computationally tractable.

## H Empirical tractability benchmark

The benchmark included Regular GP on empirical means, Aggregated GP, uncertain-input GP, full Wasserstein GP (WGP), projected Wasserstein ARD GP (PWA-GP), principal-component PWA (PCPWA), sliced Wasserstein GP (SWGP), KME-GP, and MMD-GP. The fixed benchmark suite reused the six synthetic scenarios: 2D-mean, 1D-EIV, 1D-var, 1D-skew, 2D-aniso-PC, and HD-Ackley-5D. In addition, we ran three scaling studies that varied the number of samples per cloud $M \in \{10, 20, 40, 80, 160\}$, the ambient dimension $d \in \{1, 2, 5, 10, 20\}$, and the number of training clouds $N \in \{10, 20, 30, 40, 60\}$. Each configuration was repeated for three random seeds, and for each run we recorded preprocessing time, one-shot kernel assembly time, hyperparameter training time, prediction time, total runtime, RMSE, empirical coverage of nominal 90% intervals, and CRPS. Training used the same exact-GP maximum-likelihood routines as the main experiments, with a common optimizer budget of 50 L-BFGS iterations.

Across the full benchmark, we obtained 675 successful runs and 27 benchmark-induced skips. The skips were concentrated in Aggregated GP (24 configurations) once the effective sample-level training size exceeded the benchmark cap, and in SWGP (3 configurations) for the highest-dimensional scaling point. These skips are informative rather than incidental: they expose the same tractability limits suggested by Table G.1.

We also imposed two pragmatic safeguards to keep the benchmark stable. First, Aggregated GP was skipped once the effective sample-level training size exceeded 1500 points, since its pointwise GP formulation scales with the expanded design size $NM$. Second, SWGP was not run beyond dimension $d = 10$, where the reference implementation became too costly and numerically fragile. These skips are informative rather than incidental: they reflect the same tractability issues suggested by the asymptotic costs in Table G.1.

The resulting timings show that, in this reference Python implementation, training time dominates total runtime for all exact distributional GP variants. On the fixed six-scenario suite, the median total runtimes were 0.043 s for Regular GP, 3.676 s for uncertain-input GP, 4.947 s for KME-GP, 12.905 s for MMD-GP, 33.699 s for WGP, 43.025 s for PWA-GP, and 47.810 s for PCPWA. Aggregated GP had a median total runtime of 39.237 s on the subset of runs where it remained feasible, while SWGP had a lower median runtime (8.143 s) but was numerically unstable in several scenarios and therefore is not a reliable practical alternative. Thus, among the numerically stable OT-based exact GPs in our current implementation, full WGP was the fastest, followed by PWA-GP and PCPWA.

The scaling experiments sharpen this picture. In the 2D anisotropic rotated setting, increasing the cloud size from $M = 10$ to $M = 160$ increased median total runtime from 23.3 s to 222.0 s for WGP, from 46.0 s to 293.7 s for PWA-GP, and from 51.9 s to 341.3 s for PCPWA. In the 10D Ackley scaling experiment, the same increase in $M$ changed total runtime from 31.7 s to 256.9 s for WGP, from 143.5 s to 2338.6 s for PWA-GP, and from 233.7 s to 1705.2 s for PCPWA. Similarly, when the number of training clouds in the 2D anisotropic setting increased from $N = 10$ to $N = 60$, the median total runtime rose from 3.5 s to 109.1 s for WGP, from

6.3 s to 196.1 s for PWA-GP, and from 6.5 s to 246.4 s for PCPWA. These measurements confirm that all exact distributional GP models become expensive as either the number of clouds or the cloud size grows.

At the same time, the empirical timings also show an important implementation-level nuance. Table G.1 predicts a more favorable asymptotic dependence on $M$ for PWA-GP and PCPWA than for full WGP, because the projected kernels reduce each pairwise comparison to repeated one-dimensional Wasserstein computations rather than a full multivariate OT solve. However, that asymptotic advantage does not translate into lower end-to-end wall-clock time in the present pure-Python reference implementation. In our code, the projected kernels incur substantial constant factors from repeated per-dimension sorts, repeated kernel reconstruction during L-BFGS hyperparameter optimization, and (for PCPWA) the projection overhead. We therefore view Table G.1 and the runtime benchmark as complementary: the table characterizes the kernel formulas asymptotically, while the empirical timings describe the behavior of the current unoptimized implementation.

| Method | Kernel input object | Dist. | Latent $X$ | Det. | EIV mechanism | Citations |
|---|---|---|---|---|---|---|
| **Regular GP** (RBF / Matérn / Exp) | Point estimate (empirical mean $m_i$) | N | N | Y | Ignores input uncertainty; Sec. 3/Table 1 baseline. | Williams and Rasmussen (2006) |
| **Aggregated Regular GPs** | Noisy samples $\{U_{ij}\}$ as point inputs | N | N | Y | Averages replicate-specific GPs; may inflate sparse-region variance. | **This paper.** |
| **IV-function / cov-only encoding** | Gaussian summary $(\mu_x, \Sigma_x)$ but only $\Sigma_x$ used in kernel | Y* | N | Y | Fast for local Gaussian noise; misses skew, multimodality, complex support. | Moreno-Muñoz et al. (2018) |
| **WGP** (Eq. (5), WGP-RBF) | Measure $\mu$ | Y | N | Y | OT kernel on distributions; PD in special cases (Gaussian $p=2$ or 1D). | Candelieri et al. (2022); Peyré et al. (2019) |
| **sliced WGP** (Eq. (6), SWGP) | Measure $\mu$; average $W_p$ over random 1D projections $u$ | Y | N | MC | Projection expectation preserves PD; random slices add MC cost/variance. | Meunier et al. (2022); Bonet et al. (2023) |
| **PWA-GP** (Eq. (7)) | Coordinate-wise 1D marginals $\{\mu_i\}_{i=1}^d$ | Y | N | Y | Product 1D Wasserstein kernels with ARD scales; PD by product closure. | **This paper.** |
| **PCPWA** (Eq. (8)) | Fixed directions $\{v_r\}_{r=1}^m$ 1D pushforwards | Y | N | Y | PWA on fixed/PCA directions; $m$ controls cost–expressivity. | **This paper.** |
| **Latent-input calibration GP** | Noisy $U$ with latent true $X$ (integrate out) | N | Y | (inf.) | Latent-variable EIV; requires integration/sampling approximations. | Kennedy and O'Hagan (2001) |
| **Uncertain-input GP** | Input distribution (often Gaussian) used in prediction / propagation | Y | (approx) | Y | Propagates Gaussian uncertain regressors via approximations/closed forms. | Girard et al. (2002) |
| **GP training w/ input noise** | Noisy inputs + local approximation | N | (approx) | Y | Approximates input-noise effect in training, e.g., local linearization. | McHutchon and Rasmussen (2011) |
| **Deep GP** | Joint latent representation for inputs/outputs | N | Y | (inf.) | Flexible latent EIV model; inference is heavier/approximate. | Damianou and Lawrence (2013) |
| **Low-rank Fréchet EIV regression** | Noisy covariates w/ low-rank approximation | N | N | Y | Low-rank non-GP EIV comparator; not experimentally compared. | Song and Han (2023) |
| **KME / MMD distribution regression** | Empirical samples per input distribution | Y | N | Y | Non-OT distribution baseline using RKHS geometry; often cheaper. | Szabó et al. (2016); Muandet et al. (2017) |

Table A.1: Compact comparison of EIV / uncertain-input methods. Dist. indicates whether the model consumes a distribution/set input (Y* = Gaussian-summary input but only covariance used). Det. denotes deterministic vs Monte Carlo (MC). OT = optimal transport; PD = positive definite; KME = kernel mean embedding; MMD = maximum mean discrepancy; PCA = principal component analysis.

| Method | Kernel/model reference | Comment |
|---|---|---|
| Regular GP | Standard Euclidean kernel on empirical means | Ignores input uncertainty beyond the mean representation. |
| Uncertain-input GP | Gaussian input-noise model | Uses Gaussian summaries of input uncertainty. |
| WGP | (4) | Full Wasserstein kernel on probability measures. |
| SWGP | (5) | Sliced Wasserstein kernel with random projections. |
| PWA-GP | (6) | Deterministic projected Wasserstein ARD kernel. |
| PCPWA | (7) | Fixed-basis/principal-component projected variant. |
| KME/MMD-GP | Kernel-mean or MMD distributional kernels | Non-OT distributional baselines. |

Table A.2: Compact comparison of GP methods for input measurement uncertainty. Equation references use labels from the main text to avoid hard-coded numbering drift.

| Method | Effective # inputs | Kernel assembly cost (big-O) |
|---|---|---|
| Regular GP on means (Williams and Rasmussen, 2006) | $N$ | $\mathcal{O}(N^2 d)$ |
| Aggregated GP on samples (Williams and Rasmussen, 2006) | $NM$ | $\mathcal{O}(N^2 M^2 d)$ (matrix) $+ \mathcal{O}(N^3 M^3)$ (training) |
| Uncertain-input GP (Girard et al., 2002) (McHutchon and Rasmussen, 2011) | $N$ | $\mathcal{O}(N^2 d^3)$ (full cov.) or $\mathcal{O}(N^2 d)$ (diag) |
| Full WGP (Mallasto and Feragen, 2017) (Bachoc et al., 2017) (Candelieri et al., 2022) | $N$ | $\mathcal{O}(N^2 M^2 d)$ to $\mathcal{O}(N^2 M^3)$ |
| PWA-GP (separable Wasserstein) | $N$ | $\mathcal{O}(N^2 d M \log M)$ |
| PCPWA (PCA-based PWA) | $N$ | $\mathcal{O}(N M d^2)$ (PCA) $+ \mathcal{O}(N^2 k M \log M)$ |
| SWGP (sliced Wasserstein) (Meunier et al., 2022) | $N$ | $\mathcal{O}(N R M d) + \mathcal{O}(N^2 R M \log M)$ |
| KME-GP (Muandet et al., 2017) | $N$ | $\mathcal{O}(N^2 M^2 d)$ |
| MMD-GP (Szabó et al., 2016) | $N$ | $\mathcal{O}(N^2 M^2 d)$ |
| Low-rank Fréchet EIV regression (Song and Han, 2023) | $N$ | $\mathcal{O}(N M r) + \mathcal{O}(N^2 r)$ |

Table G.1: Leading-order cost of assembling the $N \times N$ kernel matrix for the different methods, assuming $m_i \asymp M$ samples per cloud. All methods then incur a shared $\mathcal{O}(N^3)$ cost for exact GP training.

| Method | Kernel (s) | Train (s) | Predict (s) | Total (s) | RMSE | Coverage | CRPS |
|---|---|---|---|---|---|---|---|
| Regular GP | 0.000 | 0.031 | 0.004 | 0.043 | 0.229 | 0.558 | 0.128 |
| Uncertain-input GP | 0.047 | 3.35 | 0.274 | 3.68 | 0.278 | 0.567 | 0.171 |
| KME-GP | 0.038 | 4.58 | 0.308 | 4.95 | 0.350 | 0.600 | 0.212 |
| MMD-GP | 0.153 | 11.71 | 0.702 | 12.90 | 0.188 | 0.754 | 0.105 |
| WGP | 0.289 | 32.24 | 1.30 | 33.70 | 0.190 | 0.992 | 0.106 |
| PWA-GP | 0.328 | 41.11 | 1.58 | 43.02 | 0.190 | 0.971 | 0.104 |
| PCPWA | 0.335 | 45.76 | 1.62 | 47.81 | 0.195 | 0.975 | 0.114 |
| SWGP | 0.211 | 6.15 | 0.875 | 8.14 | 1.42e+06 | 0.467 | 1.15e+06 |
| Aggregated GP | 0.027 | 22.29 | 17.59 | 39.24 | 0.524 | 0.029 | 0.441 |

Table H.1: Median wall-clock time and predictive metrics on the fixed six-scenario benchmark suite. All values are medians over successful runs across the three benchmark seeds. The final column reports successful runs over configured runs; Aggregated GP was skipped on some configurations once the expanded design size exceeded the benchmark cap, and SWGP is shown separately despite severe numerical instability in its predictive metrics.

