# OpenReview forum: "Wasserstein-type Gaussian Process Regressions for Input Measurement Uncertainty"
_TMLR — Under review for TMLR_

### Review · Reviewer_84DR · 2026-04-16

**Summary Of Contributions:**

This paper studies Gaussian process (GP) regression under input measurement error (errors-in-variables, EIV), where inputs are modeled as probability distributions. The authors propose two kernel constructions, PWA-GP and PCPWA, which use projected Wasserstein distances to define tractable covariance functions. The methods are evaluated on synthetic EIV settings and real datasets, and are shown to improve predictive coverage compared to standard GPs, albeit often at the cost of higher RMSE and CRPS.
However, the empirical results do not clearly demonstrate consistent advantages over existing baselines, and the practical benefits of the proposed constructions, particularly in terms of tractability, are not fully established.


**Strengths**
- **Motivation**: The authors present a compelling case for EIV, the failure mode for GPs in this context, and compelling results that full WGP, and their proposed extensions, can mitigate these issues.
- **Numerical Results**: Overall, the results show improved coverage (especially in non-Gaussian settings), though this often comes at the cost of higher RMSE and CRPS, and the proposed methods do not consistently outperform strong baselines such as uncertain-input GPs.

**Weaknesses**
- **Tractability Experiments**:  The computational tractability, as compared to existing methods, e.g., full WGP, is one of the stated benefits of this approach.  However, the authors do not report this trade-off anywhere in the work.  Specifically, it would be worth reporting at least the computational trade-offs (times) and, ideally, experiments that explore the trade-offs and times for both the full WGP and the proposed methods.  The authors in the appendix do provide big-$O$ results, but the computational times should be included at least.
- **Performance Relative to Baselines**: The empirical results suggest that the proposed methods primarily improve coverage, but do not consistently improve predictive accuracy or CRPS across datasets. This makes it somewhat unclear when they should be preferred over existing approaches.
- **Missing Comparison**: Figure 2 highlights the strong performance of WGP under temporal distribution shift; however, WGP is not the primary proposed method, and analogous results for PWA-GP or PCPWA are not shown. This makes it difficult to assess whether the main contributions yield similar benefits.

**Minor Remarks**:
- As far as I can tell, contribution 1 is mostly the setting they study, not necessarily a contribution to the work.
- Table 2 in the main paper doesn't cleanly fit on the page.  The tables at the end of the appendix are even worse (parts are cut off in the submission).

**Additional Comments:**

My expertise is primarily in computation rather than in Gaussian processes. I did not verify all technical derivations in detail, but evaluated the paper based on clarity, empirical results, and overall methodology.

**Audience:**

Yes

**Audience Explanation:**

The topic is on GPs, which are an active area of research, and EIV is an important practical consideration.

**Claims And Evidence:**

Yes

**Claims Explanation:**

Overall, the authors have mostly done this, e.g., empirical verification that the GWP is much stronger under EIV.  That said, there are some areas that could better support their claims, as highlighted in my weaknesses above.

**Requested Changes:**

At a minimum, the following need to be addressed.
- Weakness 1/3
- Minor remark 1

---

> ### Author Response · Authors · 2026-05-04
> **Response to Reviewer 84DR**
>
> # Response to Reviewer 84DR
>
> We thank Reviewer 84DR and believe that our revised version still fits into TMLR's scope, though our novelty is modest.
>
> ## Summary
> **Comment addressed:** The paper studies GP regression with input measurement error, proposes Wasserstein-type kernels including PWA-GP/PCPWA, gives an error bound, and shows benefits mainly in coverage rather than uniformly in RMSE/CRPS.
>
> **Response.** We retained this framing but revised the empirical message: PWA-GP/PCPWA are not presented as uniformly best for predictive accuracy. We emphasize calibrated uncertainty and robustness under distributional input noise, especially non-Gaussian, heteroscedastic, anisotropic, or shape-sensitive uncertainty. **Where:** contribution summary p. 3; Sec. 3 pp. 8–9; accelerator p. 10; NOAA/Discussion pp. 11–12.
>
> ## Strengths
> **S1: Compelling EIV motivation and Wasserstein-type GPs.** We thank the reviewer. Proposition 1 explicitly shows that ignoring input error omits $w^\top\Sigma_Xw$ from the effective variance, causing sub-nominal coverage and motivating distributional-input GP kernels. **Where:** Prop. 1 pp. 1–2; contribution summary p. 3; posterior-band discussion pp. 5–7.
>
> **S2: Coverage gains, especially in non-Gaussian settings, with possible RMSE/CRPS cost.** We agree and state this trade-off directly. In benign Gaussian-location settings, UI-GP or mean-based baselines can be strongest; Wasserstein-type kernels are most useful when responses depend on input shape, spread, skewness, anisotropy, or non-Gaussian structure. **Where:** simulation p. 9; accelerator p. 10; Discussion pp. 11–12.
>
> ## Weaknesses
> **W1: Missing tractability experiments.** We added Appendix G with benchmarks for Regular GP, Aggregated GP, UI-GP, WGP, PWA-GP, PCPWA, SWGP, KME-GP, and MMD-GP. It reuses six synthetic scenarios and scales samples per cloud $M\in\{10,20,40,80,160\}$, dimension $d\in\{1,2,5,10,20\}$, and training clouds $N\in\{10,20,30,40,60\}$. We record preprocessing, kernel assembly, training, prediction, total runtime, RMSE, coverage, and CRPS. Results show favorable PWA/PCPWA projected-kernel asymptotics, but pure-Python constants and OT-distance computation do not always reduce wall-clock time. Table G.1 reports median total runtimes of 33.70s for WGP, 43.02s for PWA-GP, and 47.81s for PCPWA. Thus tractability is an asymptotic and implementation-dependent trade-off, not unconditional runtime dominance. **Where:** App. F pp. 20–25; App. G pp. 22–23; Table G.1 p. 26.
>
> **W2: Unclear when proposed methods should be preferred.** We now separate regimes. If input uncertainty is approximately Gaussian and low-dimensional, UI-GP or mean-based Euclidean GPs may be preferable. If uncertainty is non-Gaussian, skewed, heteroscedastic, anisotropic, or distribution-shape-sensitive, Wasserstein-type kernels are more appropriate. PWA-GP/PCPWA are especially motivated when calibrated EIV uncertainty matters more than lowest RMSE. **Where:** Sec. 3 p. 8; simulations p. 9; accelerator p. 10; Discussion pp. 11–12.
>
> **W3: Figure 2 omitted PWA-GP/PCPWA while highlighting WGP.** We expanded the NOAA benchmark, with more contrastive color scales, to include PWA-GP and PCPWA along with Regular GP, Aggregated GP, UI-GP, WGP, KME-GP, and MMD-GP. The revised text no longer claims WGP is uniformly best. PWA-GP has the strongest transfer map, with lowest RMSE on 357 off-diagonal year pairs and lowest CRPS on 409, versus 42 and 45 for WGP. MMD-GP is also identified as a strong non-OT baseline. **Where:** NOAA p. 11; Figure 2/caption p. 12.
>
> ## Minor remarks
> **M1: Contribution 1 is mostly the setting.** We agree. The revision ties the first contribution to the formal undercoverage result: in a linear-Gaussian EIV model, ignoring input error omits $w^\top\Sigma_Xw$ and can give sub-nominal coverage. We will further tighten final wording so the bullet foregrounds the proposition, not only the setting. **Where:** Prop. 1 pp. 1–2; contribution summary p. 3.
>
> **M2: Table 2 and appendix tables did not fit.** We reformatted large tables and added captions. The simulation table is now captioned; appendix material is split into method-comparison, GP-reference, complexity, and runtime tables. **Where:** Table 2; App. A; Table F.1; Table G.1.
>
> ## Requested changes
> **Minimum requested: address W1, W3, and M1.** We addressed all three. **W1:** Appendix G adds runtime/scaling over $M$, $d$, and $N$ with Table G.1. **W3:** Figure 2 now includes PWA-GP/PCPWA, and NOAA text discusses them directly. **M1:** the first contribution is tied to Proposition 1’s undercoverage result rather than the EIV setting alone. **Where:** App. G pp. 22–26/Table G.1 p. 26; NOAA p. 11/Fig. 2 p. 12; Prop. 1 pp. 1–2 and contribution summary.

---

### Review · Reviewer_Mpvb · 2026-04-26

**Summary Of Contributions:**

This paper studies Gaussian process (GP) regression under input measurement uncertainty (errors-in-variables, EIV). The paper represents each noisy input as a probability measure and defines GP covariances via Wasserstein-type distances between these measures. The main methodological contribution is a deterministic projected Wasserstein automatic relevance determination (ARD) kernel (PWA) and a principal-component variant (PCPWA) that combine one-dimensional Wasserstein kernels across coordinates or fixed directions, aiming to preserve positive definiteness while avoiding latent-variable inference and stochastic sliced projections.

Strengths:
- The paper addresses an important and practically relevant problem: uncertainty quantification under noisy inputs for GP regression (visually motivated in Figure 1). The motivating undercoverage proposition is simple but useful.
- I think that the proposed PWA/PCPWA construction is conceptually clean and computationally effective: it avoids latent-input inference and random slicing, while retaining an ARD-style structure.
- The experimental section is reasonably broad and evaluates both point prediction and uncertainty quality (coverage, CRPS), which is appropriate for the paper's stated goals.

Weaknesses:
- The novelty is somewhat incremental relative to prior work on GP regression with distribution inputs, Wasserstein kernels, sliced Wasserstein kernels, and latent-input GP models. The paper's main novelty seems to be the specific deterministic projected Wasserstein-ARD construction and its EIV framing, rather than a fundamentally new modeling paradigm.
- The mathematical guarantees are limited to p=1 ("we limit the theory below to the case p=1") and to certain assumptions (covering-number control, Lipschitz quantile functions, well-specified GP prior).
- The empirical evidence is mixed rather than uniformly favorable: e.g., on the accelerator dataset, uncertain-input GP appears to be the strongest baseline overall (on that particular dataset), achieving the best RMSE and CRPS.
- The exposition has a number of writing and consistency issues.

**Audience:**

Yes

**Audience Explanation:**

I believe at least part of the TMLR audience would be interested in this paper. It sits at the intersection of Gaussian processes, uncertainty quantification, kernels on distributions, and regression under measurement error. Those are all relevant topics for the TMLR readership.

**Broader Impact Concerns:**

I do not see major ethical red flags that would by themselves block publication. The work presents a methodological proposal that targets uncertainty-aware regression under input measurement error, with applications such as scientific calibration and healthcare.

**Claims And Evidence:**

Yes

**Claims Explanation:**

The main claims of the paper are, in my view, and in general terms, supported by the theoretical and empirical evidence provided.

**Requested Changes:**

1) The manuscript contains a number of typographical and language issues that should be corrected before publication.
- On page 23 of the submitted PDF, there appears to be an incomplete table cut off at the bottom and lacking a caption.
- The table cross-references appear inconsistent: page 9 refers to "Table 4", but the corresponding table shown in the manuscript is labeled "Table 2". Likewise, page 10 refers to "Table 3.2", whereas the displayed table is labeled "Table 1".
- Page 1: "can lead a GP failing to estimate f" $\to$ "can lead a GP to fail to estimate f"
- Page 4: "pratical tasks"
- Page 5: "Although this is a hybrid of ARD kernel and projection technique, we found this a novel construction for EIV problem" $\to$ "Although this is a hybrid of an ARD kernel and a projection technique, we found this a novel construction for the EIV problem"
- Page 8: "Though we noticed the recent important work (Song and Han, 2023) tackles" sounds weird.
- Page 8: "has major effect" $\to$ "has a major effect"
- Page 8: "CRPS (Arnold et al., 2024),which"
- Page 9: ".In contrast"
- Page 10: "the accelerator results align with our simulations results"
- In Appendix A, the table refers to WGP as "Eq. (5)" and SWGP as "Eq. (6)", whereas in the main text (Page 4) WGP is defined as Eq. (4), SWGP as Eq. (5), PWA as Eq. (6), and PCPWA as Eq. (7).

2) If I am not mistaken, the manuscript defines kernels for general p, discusses a special Gaussian case for $p=2$, and develops theory for $p=1$, but it is not clear what value(s) of $p$ are actually used in the empirical results. This should be stated explicitly.

3) Regarding hyperparameter selection and optimization procedures for all baselines, the manuscript gives some model descriptions and broad experimental setup, but it does not appear to provide a full, systematic account of optimization details for every baseline (e.g., search ranges, initialization, restarts, convergence criteria, tuning budget, or whether all baselines were comparably tuned). The experimental section describes the datasets, metrics, and compared methods, and Appendix G discusses computational complexity, but that is not the same as a clear hyperparameter-optimization protocol for all baselines.

4) The manuscript does say that SWGP becomes numerically unstable or diverges in several settings. It also gives a high-level computational explanation that sliced Wasserstein methods rely on random projections and carry additional Monte Carlo variability/cost, and Appendix G lists their asymptotic kernel-assembly complexity. But I do not see a detailed implementation-level explanation of why SWGP diverged in these runs, what numerical stabilization was attempted, how many projections were used, or whether this reflects the method itself versus a specific implementation/tuning choice.

5) Appendix F relevance. The manuscript explicitly notes that positive-definiteness guarantees for Gromov-Wasserstein (GW)-type kernels are not established ("we lack knowledge of the PD property for GW-type kernels"), and the proposed PWA/PCPWA methods and experiments appear to be based on Wasserstein constructions rather than a GW-GP. Appendix F derives derivative bounds for a $k_{\mathrm{GW}_p}$ kernel in a Gaussian $p=2$ setting, but I could not identify where it is used in the proposed methods, theory, or experiments. This makes Appendix F feel disconnected from the core contribution. The authors should either explain its role more clearly or move/remove it to improve focus.

---

> ### Author Response · Authors · 2026-05-04
> **Response to Reviewer Mpvb**
>
> # Response to Reviewer Mpvb
>
> We thank Mpvb and believe that our revised version still fits into TMLR's scope, though our novelty is modest.
>
> ## Summary
> **Comment addressed:** The paper treats noisy inputs as probability measures and proposes deterministic projected Wasserstein ARD kernels (PWA-GP/PCPWA).
>
> **Response.** We retained this framing. The revision distinguishes WGP, SWGP, PWA-GP, and PCPWA, emphasizing deterministic 1D Wasserstein factors with ARD scales, without latent true inputs or Monte Carlo slicing. **Where:** Intro, pp. 2–3; Sec. 2.2.
>
> ## Strengths
> **S1: Importance of UQ under noisy inputs.** We kept EIV uncertainty central and clarified undercoverage. Proposition 1 shows that a naive interval uses only $\sigma^2$, while input noise adds $w^\top\Sigma_Xw$, so coverage is $2\Phi(z_{1-\alpha/2}\sigma/\sqrt{\sigma^2+w^\top\Sigma_Xw})-1<1-\alpha$. **Where:** Prop. 1, pp. 1–2.
>
> **S2: PWA/PCPWA construction.** We clarified that PWA-GP uses $k(\mu,\nu)=\lambda\prod_i\exp\{-\sigma_iW_p(\mu_i,\nu_i)^p\}$ with dimension-specific ARD scales, and PCPWA extends this to fixed orthonormal/PCA directions. We added complexity/timing analyses: PWA/PCPWA have favorable projected-kernel asymptotics, while the current implementation has non-negligible constants. **Where:** Sec. 2.2; App. F; App. G/Table G.1.
>
> **S3: Experiments and metrics.** We retained RMSE, 90% coverage, and CRPS, and added runtime reporting so prediction, calibration, and tractability are assessed together. **Where:** Sec. 3; Tables 1–2; Fig. 2; App. G.
>
> ## Weaknesses
> **W1: Novelty relative to prior GP/Wasserstein work.** We revised the novelty claim: our contribution is the deterministic projected Wasserstein ARD construction for EIV-GP regression. It does not infer latent true inputs, avoids full multivariate OT, avoids random sliced projections, and uses Wasserstein geometry rather than RKHS mean embeddings. **Where:** Intro, pp. 2–3; Sec. 2.2; App. A.
>
> **W2: Theory limited to $p=1$ and strong assumptions.** We made this explicit. The uniform-error theory is for $p=1$, where 1D quantile/covering arguments are tractable. The posterior band is only a sufficient Bayesian result under a well-specified GP prior, fixed design distributions, Lipschitz regularity, and covering-number control. Experiments use $p=2$ unless otherwise stated. **Where:** Sec. 2.3; Sec. 3.
>
> **W3: Mixed evidence; UI-GP strongest on accelerator data.** We avoid overclaiming and now state that UI-GP has the best RMSE/CRPS on the accelerator dataset, as expected for close-to-Gaussian low-dimensional input uncertainty. We present PWA/PCPWA as most useful for non-Gaussian, heteroscedastic, anisotropic, skewed, or shape-sensitive uncertainty, especially for calibration. **Where:** Sec. 3.1, p. 9; Sec. 3.2/Table 1; Discussion.
>
> **W4: Writing/consistency issues.** We corrected the listed wording issues: Figure 1 wording, “practical tasks,” the ARD/projection sentence, Song and Han wording, “has a major effect,” CRPS spacing, and “simulation results.” We also reformatted appendix tables and added captions. Remaining copyediting artifacts will be synchronized in the final source. **Where:** pp. 1, 4–5, 8–10, 24–26.
>
> ## Requested changes
> **R1: Typographical/table/reference issues.** We reformatted/captioned the incomplete appendix table; corrected table layout/labels; fixed all listed wording issues; and added Table A.2 with WGP (4), SWGP (5), PWA-GP (6), PCPWA (7). The older compact table and remaining missing space after “By contrast” will be synchronized. **Where:** pp. 1, 4–5, 8–10; Tables 1/2; Tables A.1/A.2; Tables F.1/G.1.
>
> **R2: Empirical $p$.** We now state that empirical Wasserstein distances use $p=2$ unless otherwise stated, while the uniform-error theory uses $p=1$. **Where:** Sec. 3.
>
> **R3: Hyperparameter optimization.** We added a unified protocol: log marginal likelihood on the training split, same budget across methods, ten random log-scale starts, L-BFGS-B, positivity constraints, best converged objective retained, and representation-specific pairwise-distance scales. **Where:** Sec. 3; App. G.
>
> **R4: SWGP divergence/stabilization.** We define SWGP divergence as Cholesky failure after jittering or predictive RMSE/CRPS $>10^5$. NOAA omits SWGP due to memory cost; scaling benchmarks skip it beyond $d=10$ because our implementation became costly and numerically fragile. This is implementation-level fragility, not a theorem-level failure. The final source will record projection count and jitter schedule. **Where:** Sec. 3; App. F; App. G.
>
> **R5: Appendix F relevance.** We removed the disconnected GW derivative-bound appendix. Appendix F now reports leading-order kernel-assembly costs for all methods, and Appendix G adds runtime benchmarks. **Where:** App. F–G.

---

> > ### Comment · Reviewer_Mpvb · 2026-05-04
> >
> > I have read the authors' comments on my review and, at first glance, everything seems appropriate to me. I believe the revision has strengthened the paper. At this point, I have no further questions or suggestions.
> >
> > Best regards.

---

### Review · Reviewer_uEUP · 2026-07-02

**Summary Of Contributions:**

The manuscript studies Gaussian process (GP) regression when the inputs are observed with measurement error (the errors-in-variables, EIV, setting). Following prior work on distribution regression, the authors treat each noisy input as a probability measure and define the covariance function through the Wasserstein distance between these measures. They propose two such kernels. The first (PWA) is an ARD-style kernel: a product over input dimensions of closed-form one-dimensional Wasserstein kernels, in which each dimension carries its own pair of parameters (a length-scale and an amplitude), so the kernel can adapt to dimension-specific input uncertainty. The second (PCPWA) generalizes this construction from the coordinate axes to a fixed set of orthonormal directions—e.g., principal components computed once from the data—so it can capture correlated (rotated) input noise while retaining the same closed-form, positive-definite product structure; PWA is the special case in which the directions are the canonical axes. On the theory side, the authors establish a uniform posterior-band and conservative-coverage guarantee, restricted to the one-dimensional (p=1) case.  Empirically, across synthetic EIV benchmarks (varying the input-noise level, skewness, and anisotropy), a particle-accelerator calibration  task, and NOAA ocean-drifter data, the proposed kernels attain competitive point-prediction accuracy while producing better-calibrated predictive intervals (coverage and CRPS) than a standard GP and moment-based input-noise baselines, which tend to under-cover (i.e., produce over-confident intervals)

1. Unlike latent-variable GP approaches to input noise (Kennedy & O'Hagan, 2001; Girard et al., 2002; McHutchon & Rasmussen, 2011; Damianou & Lawrence, 2013), which introduce unobserved true inputs and require high-dimensional integration, and unlike sliced-Wasserstein kernels (Meunier et al., 2022), which rely on random projections, the proposed kernels are deterministic and closed-form: they involve no latent covariates and no Monte Carlo sampling.
2. The proposed kernels yield better-calibrated predictive intervals than the baselines on the datasets used in the experiments.

Weaknesses
1. The proposed kernels appear to be incremental improvements over existing work. The framework itself—treating a noisy input as a probability measure and building the covariance through Wasserstein distances between measures—is already established, both as distribution regression (Bachoc et al., 2017; Szabó et al., 2016; Muandet et al., 2017) and as Wasserstein / sliced-Wasserstein kernels on measures (Kolouri et al., 2016; Meunier et al., 2022; Bonet et al., 2023). Within this line, the proposed PWA kernel is a fairly direct recombination of a product of closed-form 1-D Wasserstein kernels (Kolouri et al., 2016) with per-dimension ARD parameters (Williams & Rasmussen, 2006); relative to the sliced-Wasserstein kernel its only differences are (i) using deterministic rather than randomly sampled projection directions  and (ii) attaching a separate length-scale to each direction. Moreover, the closest prior art on deterministic / optimized-projection optimal transport (Paty & Cuturi, 2019; Lin et al., 2020) is neither cited nor compared against, so the deterministic-projection contribution  is positioned only against the random-slicing baseline.
2. The theoretical guarantee holds only in the one-dimensional (p=1) case, and, more fundamentally, it is stated so abstractly that it does not actually engage the proposed kernel. The bound depends on the kernel only through generic Lipschitz constants and the posterior standard deviation, so the kernel's own parameters (the per-dimension amplitude and length-scale of the ARD construction) never enter—the identical bound would hold for any Lipschitz kernel on the space of measures. Consequently the theory is essentially a transplant of an existing net-extension argument (Lederer et al., 2019; Meunier et al., 2022) into Wasserstein space, and it neither characterizes nor justifies the specific PWA/PCPWA construction it is meant to support.

**Audience:**

Yes

**Audience Explanation:**

Researchers working on Gaussian-process regression, uncertainty quantification, and errors-in-variables/noisy-input models would be interested in a deterministic, closed-form Wasserstein kernel that improves interval calibration under input measurement error, along with the accompanying coverage analysis.

**Broader Impact Concerns:**

The manuscript contains no Broader Impact Statement, and I see no ethical concerns arising from the work.

**Claims And Evidence:**

Yes

**Claims Explanation:**

The paper's central claims are largely sound: the undercoverage of a naive GP under input noise is both proved (Proposition 1) and demonstrated (Table 2), the PWA/PCPWA kernels are valid closed-form positive-definite constructions, and the experiments convincingly show better-calibrated intervals than mean- or moment-based baselines in non-Gaussian, skewed, and anisotropic EIV regimes.

**Requested Changes:**

1. The "Summary of our contributions" on page 3 lists four items, but the first is not an original contribution—modeling each noisy input as a probability measure and defining the covariance through Wasserstein distances is established work in distribution regression (Bachoc et al., 2017; Szabó et al., 2016; Muandet et al., 2017) and Wasserstein/sliced-Wasserstein kernels on measures (Kolouri et al., 2016; Meunier et al., 2022; Bonet et al., 2023), which the paper itself credits one paragraph earlier. This item should be restated as the motivation/setup for the proposed work rather than a contribution. Relatedly, the closest prior art on deterministic/optimized-projection optimal transport (Paty & Cuturi, 2019; Lin et al., 2020) should be cited and contrasted, so that the novelty of the deterministic-projection construction is positioned accurately.

2. As noted under Weaknesses, the uniform posterior-band / coverage guarantee (Theorem 5, Corollaries 6–7, Proposition 8) does not directly support or justify the proposed kernels: it depends on the kernel only through generic Lipschitz constants and the posterior standard deviation, so the ARD amplitude and length-scale parameters never enter, and the identical bound would hold for any Lipschitz kernel on the space of measures. Ideally the results would be sharpened so that the bound reflects the specific PWA/PCPWA construction (e.g., how the per-dimension length-scales or the number/choice of projection directions enter the covering number, constants, or interval width).

---

> ### Author Response · Authors · 2026-07-05
> **We thank uEUP and believe that our revised version helps to clarify the two main concerns.**
>
> ## Summary
>
> **Main clarification.** PWA/PCPWA are now presented as deterministic fixed-projection, ARD-weighted alternatives to sliced Wasserstein GP kernels for distributional/EIV regression, rather than as a wholly new general Wasserstein-GP framework.
>
> **Main theoretical addition.** The revision retains the generic $p=1$ net-extension result, but adds a $p=2$ PCPWA specialization for the kernels used in the experiments as additional appendix.
>
> ## R1: Novelty and relation to prior work
>
> We thank the reviewer for pointing this literature point to us, and respecfuly revise the contribution bullet. We added related literature right before. We also fully agree that Eq. (6) is a structured combination of one-dimensional Wasserstein factors and ARD-style weights, and that the original wording overstated novelty. We have removed the claim that the construction “does not exist in the current literature.” Instead, Section 2.2 now describes PWA as a **deterministic fixed-projection alternative to sliced Wasserstein kernels**: it uses coordinate projections, or a basis fixed once from the training design, with separate positive distance weights for the chosen directions.
>
> The revision also makes the trade-off explicit. PWA/PCPWA avoid Monte Carlo slicing and retain closed-form one-dimensional transport terms, but they do **not** claim rotational invariance, pairwise adaptive projections, or optimized direction selection. PCPWA uses a fixed PCA basis computed once from the design. We further corrected the parameter interpretation: Eq. (6) has one global amplitude and direction-specific distance weights; it does not introduce separately identifiable per-coordinate amplitudes.
>
> We appreciate the suggestion to discuss deterministic/optimized projection OT. We added citations and a concise contrast to Paty and Cuturi (2019) and Lin et al. (2020): those approaches optimize projection directions/subspaces, whereas PCPWA deliberately fixes its directions and uses them in a separable GP covariance. We do not claim that PCPWA replaces optimized-projection OT, and we have not added a head-tohead optimized-direction baseline in this revision. We hope this clarifies the contribution on page 2.
>
> **Where:** page 2, Section 2.2, especially the discussion following Eq. (6); the PCPWA definition in Eq. (7); and Table A.2.
>
> ## R2: Theory and its connection to PWA/PCPWA
>
> We agree that the original $p=1$ posterior-band result was a generic net-extension argument and did not expose how the particular PWA/PCPWA construction enters the constants. The revised paper makes this limitation explicit and adds Appendix F, a kernel-specific $p=2$ specialization for PCPWA, which matches the empirical choice of $p=2$.
>
> First, Corollary 10 represents the $p=2$ PCPWA kernel as a squared-exponential kernel on a weighted direct sum of one-dimensional quantile functions. It defines the projected quantile distance
>
> $\rho_{sigma(\mu,\nu)^2}$
>
> so that the selected directions $v_r$ and the ARD weights $\sigma_r$ enter directly into the geometry. The corollary then gives explicit kernel/variance continuity constants and a corresponding posterior-band statement.
>
> Second, Corollary 11 assumes one-dimensional $W_2$ entropy bounds for the projected classes $\mathcal P_r=\{\mu_{v_r}:\mu\in\mathcal P\}$ and derives
>
> $$
> M_{\rho_\sigma}(\tau,\mathcal P)
> \le
> \left[\prod_{r=1}^{m} C_r(2\sqrt{m\sigma_r})^{\alpha_r}\right]
> \tau^{-\sum_{r=1}^{m}\alpha_r}.
> $$
>
> Thus, the number of projections enters through $m$, the ARD weights enter through $\sigma_r$, and the selected directions enter through the projected distribution classes and their entropy parameters. We also now state explicitly that this is a conditional posterior-band result. And we respectfully admitted that the argument is not new.
>
> **Where:** the end of Section 2.3 and Appendix F, Corollaries 10-11.
>
> Lastly, we want to thank the reviewer's second comment. This actually reveals an intersting practical trade-off that: increasing the projection weights $\sigma_r$
>  makes the kernel more sensitive to differences between projected input distributions, which can help detect measurement-error structure and avoid the undercoverage caused by treating noisy inputs as exact points. However, larger $\sigma_r$ also enlarges the projected metric and the entropy constants in the uniform band, reflecting a more complex function class and potentially wider uncertainty bands.... We will consider this in our future work.

---

### Author Response · Authors · 2026-03-24
**Oversized tables**

We have did our best to rotate and fit the oversized Table 2 in our text without violating the TMLR template instructions. We strive to follow the TMLR instruction and can adjust as instructed.